# Dynamics-Aligned Diffusion Planning for Offline RL: A Unified Framework with Forward and Inverse Guidance

**Zihao Wang**[*]                                                   *zih.wang@nuaa.edu.cn*
**Ke Jiang**[*]                                                     *ke_jiang@nuaa.edu.cn*
**Xiaoyang Tan**[†]                                                 *x.tan@nuaa.edu.cn*
*College of Computer Science and Technology,*
*Nanjing University of Aeronautics and Astronautics,*
*MIIT Key Laboratory of Pattern Analysis and Machine Intelligence, Nanjing, China*

**Reviewed on OpenReview:** *https://openreview.net/forum?id=h3hG6EuqU2*

## Abstract

Diffusion-based planning has emerged as a powerful paradigm for offline reinforcement learning (RL). However, existing approaches often overlook the physical constraints imposed by real-world dynamics, resulting in dynamics inconsistencya mismatch between diffusion-generated trajectories and those feasible under true environment transitions. To address this issue, we propose Dynamics-Aligned Diffusion Planning (DADP), a unified framework that explicitly enforces dynamics consistency during the diffusion denoising process. DADP offers two complementary variants: DADP-F (Forward), which employs a forward dynamics model to ensure state-level feasibility, and DADP-I (Inverse), which leverages an inverse dynamics model to enhance action-level executability. Both variants share a unified guidance formulation that integrates task return optimization and dynamics alignment through gradient-based updates. Experiments on the state-based D4RL Maze2D and MuJoCo benchmarks demonstrate that DADP-F and DADP-I outperform state-of-the-art offline RL baselines, effectively reducing dynamics inconsistency and improving long-horizon robustness. This work unifies diffusion-based planning with physically grounded dynamics modeling.

## 1 Introduction

Reinforcement learning (RL) (Sutton, 2018; Silver et al., 2017) has demonstrated remarkable potential in solving complex sequential decision-making problems. However, its application in critical domains such as medical development (Fatemi et al., 2022), autonomous driving (Fang et al., 2022), and robotics control (Sinha et al., 2022) is significantly constrained by the impracticality of its trial-and-error mechanism. This limitation has spurred the development of offline RL (Kumar et al., 2020; Jiang et al., 2023; Yu et al., 2020), which focuses on learning effective policies directly from pre-collected datasets.

Since offline RL relies solely on fixed, behavior-driven datasets, most algorithms adopt conservative objectives that favor in-distribution actions. While such strategies mitigate extrapolation errors, they also induce myopic policy optimization (Jiang et al., 2023), focusing excessively on short-term rewards while neglecting the long-term consequences of early decisions. This limited foresight makes it challenging to synthesize reliable multi-step action sequencesan ability essential for modern robotic and autonomous systems. The problem is further exacerbated by perception noise, actuation uncertainty, and distributional shifts between the behavior policy and the learned policy, which lead to error accumulation and performance degradation over extended horizons.

Recent advances in diffusion-based planning have provided a probabilistic perspective on long-horizon decision-making by casting it as inference over optimal trajectory distributions. Inspired by control-as-

---

[*]Equal contribution.
[†]Corresponding author.

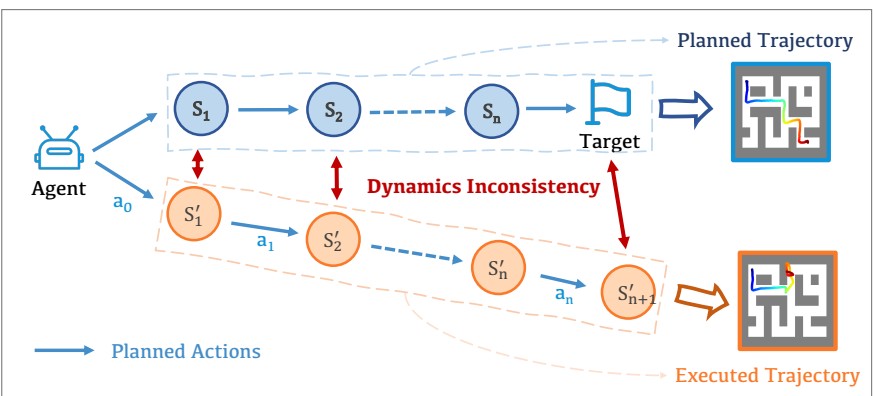

Figure 1: Illustration of dynamics inconsistency in diffusion-based planning. The diffusion model generates a planned trajectory (blue) toward the target through denoising in trajectory space. When executed in the real environment, the actual trajectory (darkgreen) deviates due to dynamics inconsistencythe mismatch between predicted and true transitions. This phenomenon motivates the proposed Dynamics-Aligned Diffusion Planning (DADP), which enforces forward (state-level) or inverse (action-level) consistency during trajectory generation.

inference (Levine, 2018) and diffusion generative models (Croitoru et al., 2023; Janner et al., 2022), these methods train diffusion models to transform Gaussian noise into complete trajectories through iterative denoising. By operating directly in trajectory space, diffusion planners can model long-range temporal dependencies without committing to step-wise policy generation.

Despite this progress, existing diffusion-based planners often fail to capture fine-grained local transition structure. Consequently, generated trajectories may exhibit global coherence yet violate local transition feasibility when executed in the real environment. We refer to this phenomenon as dynamics inconsistencysystematic mismatch between predicted and true transitions that accumulates over time (Figure 1). A key reason is that diffusion sampling is typically guided by return-related objectives and behavioral priors, while local dynamics constraints are not explicitly penalized during denoising, making errors compound over long horizons. Such inconsistencies can severely undermine reliability and robustness, particularly in complex continuous-control domains.

To address this challenge, we propose Dynamics-Aligned Diffusion Planning (DADP), a unified framework that explicitly enforces alignment between diffusion-generated trajectories and environment dynamics. DADP augments the standard denoising process with a dynamics-alignment energy term that serves as a differentiable physics-consistency constraint during reverse diffusion. At each denoising step, the model jointly optimizes task reward and local dynamic fidelity, producing trajectories that are not only high-reward but also physically realizable and dynamically stable. Conceptually, DADP transforms diffusion-based planning from pure generative inference into physics-guided optimization, bridging the gap between probabilistic modeling and real-world feasibility. Moreover, this integration enables the planner to capture richer temporal correlations and more faithful motion continuity, providing a principled way to couple learned dynamics with trajectory generation under a unified probabilistic framework.

To comprehensively address dynamics inconsistency, we further decompose it into two complementary aspects and propose two independent variants under a unified framework. DADP-F (Forward) incorporates a forward dynamics model to penalize physically invalid state transitions, ensuring that generated next states are smoothly reachable given the current states and actions within the learned dynamics manifold. In contrast, DADP-I (Inverse) employs an inverse dynamics model to correct infeasible or non-executable actions during denoising, guaranteeing that generated controls are both consistent and realizable for achieving the desired state transitions. Each variant independently enforces dynamics alignment from different perspectives: DADP-F focuses on maintaining valid and physically plausible state transitions, while DADP-I emphasizes

the executability and stability of generated actions. Together, they provide complementary insights into improving physical consistency, robustness, and long-horizon stability in diffusion-based planning.

Our main contributions are summarized as follows:

1. **Unified Framework.** We propose Dynamics-Aligned Diffusion Planning (DADP), a unified framework that explicitly integrates environment dynamics into diffusion-based trajectory generation for offline reinforcement learning.

2. **Dual Implementations.** We develop two independent variantsDADP-F (Forward) and DADP-I (Inverse)which enforce state-level and action-level consistency, respectively, addressing dynamics alignment from predictive and corrective perspectives.

3. **Principled Formulation and Mechanistic Analysis.** Unlike heuristic improvements, we derive a mathematically grounded guidance objective (Eqs. 14–16) that explicitly models dynamics inconsistency as an energy potential. We provide a mechanistic analysis demonstrating how this gradient field effectively projects diffusion trajectories onto the feasible dynamics manifold, thereby mitigating compounding errors.

4. **Comprehensive Evaluation.** Extensive experiments on D4RL Maze2D and MuJoCo benchmarks demonstrate that both variants consistently outperform state-of-the-art diffusion-based and model-based offline RL methods in terms of planning accuracy and robustness.

In this work, we focus on state-based offline RL, where the agent has access to low-dimensional proprioceptive states (e.g., joint angles, velocities). Extending our framework to high-dimensional sensory inputs (e.g., pixels) or partially observable settings remains an important direction for future research.

## 2 Related Works

**Offline Reinforcement Learning.** Offline reinforcement learning (Offline RL) aims to learn high-performing policies from static datasets while maintaining conservatism to mitigate distributional shift. Most approaches achieve this by imposing value or policy constraints that keep the learned policy within or close to the behavior distribution. Early value-based methods such as Batch-Constrained Deep Q-learning (BCQ) (Fujimoto et al., 2019) regularize the Q-target using behavior-clipped actions. Conservative Q-Learning (CQL) (Kumar et al., 2020) and Implicit Q-Learning (IQL) (Kostrikov et al., 2021) further constrain the learned value function through advantage-weighted or expectile regression objectives, effectively avoiding out-of-distribution (OOD) actions. Although these methods alleviate distributional bias, they often exhibit myopic optimization tendencies, focusing on immediate rewards and struggling to produce reliable long-horizon behaviors in complex control tasks.

**Model-Based Reinforcement Learning (MBRL).** Model-based reinforcement learning improved sample efficiency by explicitly learning environment dynamics for planning or policy optimization. Representative works such as MOPO (Yu et al., 2020) and MOReL (Kidambi et al., 2020) mitigated model bias by penalizing uncertainty or terminating rollouts that deviated from the data distribution. COMBO (Yu et al., 2021) combines model-based and value-based regularization to enhance offline policy learning. Despite their benefits, these approaches fundamentally depend on model accuracy: even small prediction errors can accumulate over time, resulting in unstable rollouts and degraded performance in high-dimensional continuous environments.

**Diffusion-Based Planning for Reinforcement Learning.** Recent advances employed diffusion models to directly generate trajectories in offline reinforcement learning. Diffuser (Janner et al., 2022) introduced a denoising diffusion probabilistic model conditioned on returns to synthesize entire stateaction sequences. Building on this idea, Decision Diffuser (Ajay et al., 2022) incorporated conditional guidance for desired outcomes, while hierarchical extensions such as HDMI (Li et al., 2023) enabled multi-level trajectory planning. Adaptive diffusion-based planning (Zhou et al., 2024) introduced uncertainty estimation for online re-planning. While these methods effectively capture global trajectory structures, they often overlook local

physical consistency, resulting in trajectories that may be statistically valid yet dynamically infeasible when executed. Our proposed Dynamics-Aligned Diffusion Planning (DADP) explicitly addresses this issue by integrating forward and inverse dynamics constraints into the denoising process, ensuring that generated trajectories remain both high-reward and physically realizable.

**Dynamics-Aware and Constrained Diffusion.** While standard diffusion planners generate trajectories in an open-loop fashion, recent works have sought to enforce dynamics constraints more explicitly. Methods such as Equality-Constrained DiffusionKurtz & Burdick (2025) and Constrained DiffusersZhang et al. (2025) typically employ optimization techniquessuch as augmented Lagrangian methods or primal-dual updatesto treat dynamics as strict equality constraints, primarily targeting direct trajectory optimization or safe planning tasks. Similarly, approaches like DPCCRömer et al. (2024) integrate continuity constraints directly into the generation pipeline via projection to enable constraint-satisfying imitation learning. While effective in their respective domains, these "hard constraint" methods often require complex inner-loop optimizations or sensitive hyperparameter tuning (e.g., Lagrange multipliers) and can be computationally intensive. Furthermore, they typically focus on satisfying constraints within expert demonstrations or analytical models, whereas our goal is to extract optimal policies from suboptimal offline data. In contrast, our DADP framework adopts a soft guidance approach via energy minimization. Instead of enforcing hard constraints that effectively project trajectories onto a potentially imperfect learned manifold, DADP leverages gradient-based guidance from learned forward/inverse VAEs. This formulation allows for a more flexible trade-off between reward maximization and physical feasibility, acts as a regularizer rather than a strict bound, and integrates seamlessly into the standard diffusion sampling loop with minimal computational overhead.

## 3 Method

### 3.1 Problem Formulation

Offline reinforcement learning (Offline RL) aims to learn optimal decision-making policies purely from pre-collected datasets, without any further interaction with the environment. The environment is modeled as a Markov Decision Process (MDP) $(\mathcal{S}, \mathcal{A}, P, r, \gamma, \rho_0)$, where $\mathcal{S}$ and $\mathcal{A}$ denote the state and action spaces, $P$ is the transition kernel, $r$ the reward function, $\gamma$ the discount factor, and $\rho_0$ the initial state distribution. An agent follows a policy $\pi : \mathcal{S} \rightarrow \mathcal{A}$ to generate trajectories $\tau = (s_0, a_0, s_1, a_1, \ldots, s_{T-1}, a_{T-1}, s_T)$ of horizon $T$. The optimal policy can be interpreted as producing the first action $a_0$ according to an optimal trajectory distribution:

$$\pi^*(a|s) = p^*(\tau|s_0 = s)(a_0), \tag{1}$$

where $p^*(\tau|s_0 = s)$ denotes the distribution over optimal trajectories starting from the initial state $s$.

In the offline RL setting, the agent has access only to a static dataset $\mathcal{D}$ with empirical trajectory distribution $P_{\mathcal{D}}(\tau)$. The learning objective becomes to infer an optimal trajectory distribution from this fixed dataset, which can be formulated as:

$$\hat{p}^*(\tau) = \arg\max_{\hat{p}} \mathbb{E}_{\tau \sim \hat{p}(\tau)}[\log P_{\mathcal{D}}(\tau)] + \mathbb{E}_{\tau \sim \hat{p}(\tau)}[\mathcal{J}(\tau)], \tag{2}$$

where $\mathcal{J}(\tau) = \sum_{t=0}^{T-1} r(s_t, a_t)$ denotes the cumulative return.

This dual-objective formulation jointly encourages distributional alignment with the offline dataset and reward maximization for optimal control. The likelihood term serves as an implicit regularizer, alleviating sparse or delayed reward issues that often arise in long-horizon planning. This formulation naturally lends itself to diffusion-based generative modeling, where the goal is to approximate the optimal trajectory distribution through iterative denoising.

### 3.2 Diffusion-based Trajectory Modeling

Diffusion probabilistic models (Sohl-Dickstein et al., 2015) are powerful generative frameworks that synthesize complex data by iteratively denoising samples through a Markovian process. In the trajectory generation setting, a diffusion model learns to map Gaussian noise to complete trajectories via a reverse diffusion

process. Specifically, the generative process is defined as a sequence of reverse transitions $p(\tau^{i-1}|\tau^i)$ that invert a forward diffusion process $q(\tau^i|\tau^{i-1})$, where the latter progressively corrupts clean trajectories by adding Gaussian noise at each step. The marginal data distribution modeled by the diffusion process can be written as:

$$p(\tau^0) = \int p(\tau^N) \prod_{i=1}^{N} p(\tau^{i-1}|\tau^i)\, d\tau^{1:N}, \tag{3}$$

where $p(\tau^N)$ is a standard Gaussian prior and $\tau^0$ denotes the clean trajectory. The reverse transitions are typically parameterized as Gaussian distributions:

$$p(\tau^{i-1}|\tau^i) = \mathcal{N}\big(\tau^{i-1}; \mu_\theta(\tau^i) + \Sigma^i \nabla_{\mu_\theta} \mathcal{J}(\mu_\theta(\tau^i)), \Sigma^i\big). \tag{4}$$

where $\mu(\tau^i, i)$ and $\Sigma(i)$ represent the mean and covariance of the reverse process, learned by a neural network conditioned on the noisy trajectory $\tau^i$ and the diffusion step $i$. The forward process $q(\tau^i|\tau^{i-1})$ is defined as a fixed Gaussian corruption kernel (Sohl-Dickstein et al., 2015).

To integrate diffusion-based generation into decision-making, we adopt the control-as-inference perspective (Levine, 2018), which formulates optimal control as a probabilistic inference problem. Let $\mathcal{O}_t$ denote a binary optimality variable at time $t$, where $p(\mathcal{O}_t = 1) = \exp(r(s_t, a_t))$. The optimal trajectory distribution can then be expressed as a posterior:

$$p^*(\tau) = p(\tau \mid \mathcal{O}_{0:T} = 1) \propto p(\tau)\, p(\mathcal{O}_{0:T} = 1|\tau). \tag{5}$$

This formulation encourages sampling trajectories that are both behaviorally plausible (high $p(\tau)$) and reward-optimal (high $p(\mathcal{O}_{0:T} = 1|\tau)$).

Following (Janner et al., 2022), when the optimality likelihood $p(\mathcal{O}_{0:T} = 1|\tau)$ is smooth, the reverse diffusion transitions can be approximated by:

$$p(\tau^{i-1}|\tau^i, \mathcal{O}_{0:T}) \approx \mathcal{N}\big(\tau^{i-1}; \mu + \Sigma \nabla_\mu \mathcal{J}(\mu), \Sigma\big), \tag{6}$$

where $\mu, \Sigma$ are parameters of the unconditional diffusion model, and $\nabla_\mu \mathcal{J}(\mu)$ represents a reward-guided gradient that steers denoising toward higher-return trajectories. This perspective bridges generative modeling and reward optimization, forming the basis for our proposed dynamics-aligned diffusion framework introduced in the next section.

### 3.3 Quantifying Dynamics Inconsistency

While diffusion-based planning can capture global trajectory structure, it may violate local transition feasibility. We quantify dynamics inconsistency in two contexts: (i) differentiable objectives used for guidance, and (ii) a simulator-based metric used for evaluation.

**Differentiable Inconsistency for Guidance**

We quantify dynamics inconsistency from two complementary perspectives using learned dynamics models. Although our dynamics models are implemented as VAEs that parameterize conditional distributions, we denote their *mean predictors* by $M_f$ and $M_i$ for notational simplicity.

**State-level inconsistency (forward).** Let $\hat{s}_{t+1} = M_f(s_t, a_t)$ be the mean prediction of the learned forward dynamics model. We define

$$E^{\mathrm{F}}(\tau) = \sum_{t=0}^{T-1} \|s_{t+1} - \hat{s}_{t+1}\|_2^2 = \sum_{t=0}^{T-1} \|s_{t+1} - M_f(s_t, a_t)\|_2^2. \tag{7}$$

**Action-level inconsistency (inverse).** Let $\hat{a}_t = M_i(s_t, s_{t+1})$ be the mean prediction of the learned inverse dynamics model. We define

$$E^{\mathrm{I}}(\tau) = \sum_{t=0}^{T-1} \|a_t - \hat{a}_t\|_2^2 = \sum_{t=0}^{T-1} \|a_t - M_i(s_t, s_{t+1})\|_2^2. \tag{8}$$

Both quantities are differentiable and are incorporated into DADP as guidance objectives to regularize dynamics during trajectory generation.

**Simulator-Based Inconsistency for Evaluation**

To assess physical realism in experiments (e.g., Figure 3b), we compute the deviation between planned and executed transitions in the ground-truth simulator:

$$E_{\text{eval}}(\tau) = \sum_{t=0}^{T-1} \|s_{t+1,\text{plan}} - s_{t+1,\text{sim}}\|_2, \tag{9}$$

where $s_{t+1,\text{sim}}$ is obtained by rolling out the planned actions in the simulator (e.g., MuJoCo). Note that $E_{\text{eval}}$ is used only for analysis and is not available during inference.

### 3.4 Learning the Dynamics Models

To enable differentiable and robust consistency guidance, both the forward and inverse dynamics models in our framework are implemented as variational autoencoders (VAEs). A probabilistic latent-variable formulation offers two main advantages over deterministic regression:

(1) It better handles the multi-modality inherent in offline datasets compared to deterministic regression, preventing the model from learning an invalid "average" transition;

(2) It learns a smooth, structured latent manifold that yields stable and informative gradients for consistency guidance.

While the VAE explicitly models variance, in this work, we primarily utilize the learned mean prediction for guidance. By leveraging the structured latent space, we ensure that the gradients remain well-behaved during optimization, focusing on prediction quality rather than explicit uncertainty weighting.

Formally, given observed transitions $(s_t, a_t, s_{t+1})$, the encoder maps inputs to a latent representation $z_\phi$, while the decoder reconstructs the conditional distribution of the target variableeither the next state or the actiondepending on the model type. Assuming a standard Gaussian prior $p(z) = \mathcal{N}(0, I)$, both models are trained by maximizing the evidence lower bound (ELBO):

$$\mathcal{L}_{\text{ELBO}} = \mathbb{E}_{q_\phi(z|x)}[\log p_\psi(y|z)] - \text{KL}(q_\phi(z|x) \,\|\, p(z)), \tag{10}$$

where $(x, y)$ denotes the inputtarget pair and $\phi, \psi$ are the parameters of the encoder and decoder, respectively. We instantiate this structure in two complementary forms:

**Forward dynamics model.** The forward dynamics model parameterizes the conditional distribution of the next state given $(s_t, a_t)$:

$$p_\psi(s_{t+1} \mid s_t, a_t) = \mathcal{N}\big(\mu_f(s_t, a_t), \Sigma_f(s_t, a_t)\big). \tag{11}$$

For guidance, we use the decoder mean and denote it as a deterministic predictor:

$$M_f(s_t, a_t) \mu_f(s_t, a_t). \tag{12}$$

In training, we minimize the mean-squared error of the mean prediction:

$$\mathcal{L}_f = \sum_{t=0}^{T-1} \|M_f(s_t, a_t) - s_{t+1}\|_2^2. \tag{13}$$

**Inverse dynamics model.** The inverse dynamics model parameterizes the conditional distribution of the action given $(s_t, s_{t+1})$:

$$p_\psi(a_t \mid s_t, s_{t+1}) = \mathcal{N}\big(\mu_i(s_t, s_{t+1}), \Sigma_i(s_t, s_{t+1})\big). \tag{14}$$

Similarly, we denote the decoder mean as

$$M_i(s_t, s_{t+1})\mu_i(s_t, s_{t+1}). \tag{15}$$

This model captures multiple feasible controls for the same transition and provides robust feasibility correction during denoising.

In practice, the forward and inverse dynamics models are used separately to support two DADP variants. DADP-F employs the forward model to guide trajectory generation toward state-level consistency, while DADP-I utilizes the inverse model to enforce action-level consistency. Each model acts as a differentiable prior that regularizes the diffusion process, ensuring that generated trajectories remain consistent with the underlying environment dynamics.

### 3.5 Dynamics-Aligned Diffusion Planning (DADP)

The proposed Dynamics-Aligned Diffusion Planning (DADP) framework integrates learned dynamics models into the reverse diffusion process to mitigate dynamics inconsistency. The key idea is to introduce a dynamics-alignment constraint during denoising, ensuring that generated trajectories remain both reward-optimal and physically feasible throughout the diffusion process.

During planning, two pretrained models are employed: a diffusion model $p_\theta(\tau)$ and a dynamics model $M$, which can be either the forward model $M_f$ or the inverse model $M_i$, depending on the DADP variant. Trajectory sampling begins by initializing $\tau^N$ from a standard Gaussian prior $\mathcal{N}(0, I)$, with the initial state fixed to the agents current observation. The reverse diffusion process iteratively denoises the trajectory over $N$ steps, where each step is represented by $p_\theta(\tau^{i-1}|\tau^i)$. At each iteration, the diffusion model outputs a mean $\mu_\theta(\tau^i)$ and covariance $\Sigma^i$ that govern the sampling distribution, ensuring both stability and diversity during denoising.

To enforce local dynamics alignment, we compute a dynamics inconsistency score at each iteration. Depending on the variant, the formulation is as follows:

**State-level alignment (DADP-F).**

$$E^{\mathrm{F}}(\tau) = \sum_{t=0}^{T-1} \gamma_t \|M_f(s_t, a_t) - s_{t+1}\|_2^2, \tag{16}$$

**Action-level alignment (DADP-I).**

$$E^{\mathrm{I}}(\tau) = \sum_{t=0}^{T-1} \gamma_t \|M_i(s_t, s_{t+1}) - a_t\|_2^2. \tag{17}$$

where $\gamma_t$ is a weighting coefficient. In our implementation, we adopt a uniform weighting scheme by setting $\gamma_t = 1$ for all timesteps. Empirical results suggest that the smooth gradients provided by the VAE latent manifold allowed for robust guidance without the need for complex, uncertainty-based adaptive weighting schedules. Thus, we prioritize algorithmic simplicity and stability, leaving adaptive weighting for future work.

At each denoising step, the trajectory is refined using gradient-based guidance that balances task rewards and dynamics consistency. The gradient of the augmented objective is computed as

$$g = \nabla_\mu(\lambda \mathcal{J} - \beta E), \tag{18}$$

where $\mathcal{J}$ represents the cumulative reward defined in Eq. 2, $\lambda$ controls the reward weighting, and $\beta$ determines the strength of dynamics alignment. The next trajectory is then sampled as

$$\tau^{i-1} \sim \mathcal{N}(\boldsymbol{\mu} + \alpha\Sigma^i g, \Sigma^i), \tag{19}$$

where $\alpha$ adjusts the influence of the guidance term while $\Sigma^i$ preserves sampling variance for exploration.

To ensure task-specific feasibility, we apply constraints $\mathcal{C}$ during denoising:

$$\mathcal{C}(\tau) = \{s_0 = s_{\text{current}}, \ s_T = s_{\text{goal}} \text{ (if specified)}\}, \tag{20}$$

where $s_{\text{current}}$ is the current state and $s_{\text{goal}}$ is an optional target. These constraints ensure that generated trajectories start from valid states and, when applicable, terminate at goal positions.

In summary, DADP extends standard diffusion-based planning by incorporating explicit dynamics alignment. The forward variant (DADP-F) promotes predictive reachability through state-level consistency, while the inverse variant (DADP-I) enforces action-level feasibility. Both independently enhance the physical validity and stability of long-horizon diffusion planning. The overall procedure is summarized in Algorithm 1.

---

**Algorithm 1** Dynamics-Aligned Diffusion Planning (DADP) for Offline Trajectory Generation

---

1: **Input:** Diffusion model $\boldsymbol{\mu}_\theta$; Dynamics model $M$ (either forward $M_f$ or inverse $M_i$); Planning horizon $T$; Number of diffusion steps $N$; Scaling factors $\alpha$, $\lambda$, $\beta$; Covariances $\{\Sigma^i\}_{i=1}^N$; Constraints $\mathcal{C}$.
2: Initialize noisy trajectory $\tau^N \sim \mathcal{N}(\mathbf{0}, \mathbf{I})$; observe initial state $s_{\text{current}}$.
3: **for** $i = N, \dots, 1$ **do**
4:     Compute denoised mean $\boldsymbol{\mu} \leftarrow \boldsymbol{\mu}_\theta(\tau^i)$.
5:     Extract state–action pairs $\{(s_t, a_t, s_{t+1})\}_{t=0}^{T-1}$ from $\boldsymbol{\mu}$.
6:     Compute cumulative reward $\mathcal{J} \leftarrow \sum_{t=0}^{T-1} r(s_t, a_t)$.
7:     Compute dynamics inconsistency term:

$$E \leftarrow \begin{cases} \sum_{t=0}^{T-1} \|M_f(s_t, a_t) - s_{t+1}\|^2, & \text{for DADP-F} \\ \sum_{t=0}^{T-1} \|M_i(s_t, s_{t+1}) - a_t\|^2, & \text{for DADP-I} \end{cases}$$

8:     Compute joint guidance gradient: $g \leftarrow \nabla_{\boldsymbol{\mu}}(\lambda \mathcal{J} - \beta E)$.
9:     Sample denoised trajectory: $\tau^{i-1} \sim \mathcal{N}(\boldsymbol{\mu} + \alpha \Sigma^i g, \Sigma^i)$.
10:     Project $\tau^{i-1}$ to satisfy constraints $\mathcal{C}$ (e.g., $s_0 = s_{\text{current}}$, optional $s_T = s_{\text{goal}}$).
11: **end for**
12: **Output:** Final planned trajectory $\tau^0$.

---

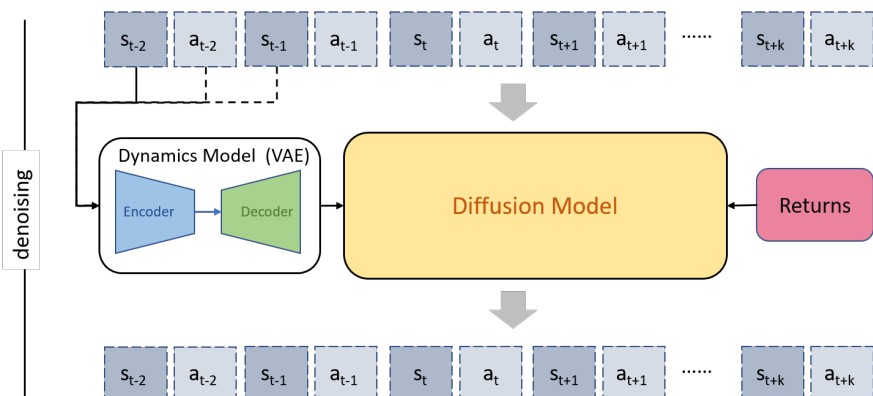

Figure 2: Overview of the proposed Dynamics-Aligned Diffusion Planning (DADP) framework. The dynamics model, implemented as a variational autoencoder (VAE), provides local physical consistency guidance during the diffusion-based trajectory denoising process. For the forward variant (DADP-F), the dynamics model predicts the next state given $(s_t, a_t)$, while for the inverse variant (DADP-I), it infers the corresponding action from $(s_t, s_{t+1})$. The diffusion model then refines the trajectory by jointly optimizing for reward maximization and dynamics alignment.

## 4 Experiments

We conduct extensive experiments to comprehensively evaluate the proposed Dynamics-Aligned Diffusion Planning (DADP) framework. The study is designed to address the following key research questions:

1. Can DADP effectively reduce dynamics inconsistency and mitigate error accumulation in long-horizon planning?

2. Do the two variants, DADP-F and DADP-I, enhance robustness and generalization across diverse continuous-control environments?

We evaluated our approach on two widely used D4RL benchmarks: Maze2D for long-horizon, sparse-reward navigation, and MuJoCo for continuous robotic control. Our implementation built upon the Diffuser codebase, employing a U-Net backbone for trajectory denoising. Both forward and inverse dynamics models are implemented as variational autoencoders (VAEs) with fully connected encoderdecoder architectures. All models are trained using the Adam optimizer, with uniform weighting ($\gamma_t = 1$). Hyperparameters $\alpha$, $\lambda$, and $\beta$ are tuned per environment via validation to ensure stable convergence.

In the following sections, we first describe the experimental setup and baseline configurations, then present quantitative results on Maze2D and MuJoCo benchmarks, followed by an in-depth analysis of the effects of dynamics alignment in DADP-F and DADP-I.

### 4.1 Maze2D: Evaluating Long-Horizon Planning

We evaluate the proposed DADP framework on the D4RL Maze2D benchmark, a suite of long-horizon navigation tasks characterized by sparse rewards. Maze2D comprises three configurationsumaze, medium, and largecorresponding to increasing maze complexity and planning horizons. The agent receives a reward of 1 only upon reaching the goal, so higher cumulative returns indicate shorter completion times within the fixed horizon. Following standard evaluation protocols, we report normalized scores, where higher values reflect more efficient planning and better task performance.

**Baselines.** We compare DADP against representative model-free and model-based offline reinforcement learning algorithms. Model-free baselines include Batch-Constrained Deep Q-Learning (BCQ) (Fujimoto et al., 2019), Conservative Q-Learning (CQL) (Kumar et al., 2020), and Implicit Q-Learning (IQL) (Kostrikov et al., 2021). Model-based methods include ROMI (Wang et al., 2021), the original Diffuser (Janner et al., 2022), and improved variants such as Diffuser-HER, Diffuser-AM (Kim et al., 2024), We also compare against Hierarchical Diffusion for Offline Decision Making (HDMI) (Li et al., 2023) and the contemporary dynamics-constrained method, DPCC (Römer et al., 2024), which enforces continuity via hard manifold projection (reproduced using the official implementation). Together, these baselines cover diverse paradigms of value regularization, model-based rollout, and diffusion-based planning.

**Results.** Table 1 reports normalized scores on Maze2D and its multi-task variant Multi2D. DADP-F and DADP-I consistently outperform all baselines across maze sizes and task settings. In the single-task scenario, DADP-F achieves an average normalized score of 137.8, surpassing Diffuser (119.5) and HDMI (123.5). DADP-I attains a similar score of 137.5, demonstrating higher robustness across random seeds. In the more challenging Maze2D-Large environment, DADP-F achieves 147.7representing a 20% improvement over Diffuser (123.0)highlighting the benefits of explicit dynamics alignment.

In the multi-task setting, where start and goal positions are randomized, DADP-F and DADP-I again yield the best overall performance, achieving averages of 144.2 and 145.9, respectively. Both variants generalize effectively to unseen configurations, while traditional diffusion-based planners show degraded performance under distributional shift.

Representative planning and execution results in Maze2D-Large are shown in Figure 3a. The first row shows planned trajectories, while the second and third rows correspond to executions under Diffuser and DADP-I, respectively. Each column depicts an independent episode. Diffuser often deviates between planned and

executed paths due to cumulative modeling errors, whereas DADP-I maintains close alignment, confirming the stability benefits of dynamics alignment.

**Comparison with DPCC.** We further evaluated the official DPCC implementation (Römer et al., 2024) on the Maze2D benchmark. As shown in Table 1, DPCC exhibits significantly lower performance compared to DADP (e.g., 50.2 vs. 130.2 in U-Maze). This performance gap primarily stems from DPCC's reliance on iterative optimization-based projection, which acts as a hard constraint. On suboptimal offline datasets, this rigid projection can force trajectories into valid but low-reward regions of the dynamics manifold, limiting the planner's ability to stitch optimal paths. In contrast, DADP employs a principled formulation based on soft, gradient-based energy minimization. This approach gracefully balances reward maximization with physical feasibility, allowing the planner to effectively "stitch" high-reward segments while penalizing dynamics violations. The advantage is evident in the complex Maze2D-Large task, where DADP-F achieves 147.7 compared to DPCC's 96.5, highlighting the robustness of our framework.

Figure 3b further provides a quantitative comparison of dynamics inconsistency (evaluated against the ground-truth simulator as defined in Eq.9) across representative episodes. Diffuser exhibits rapidly increasing inconsistency over time, indicating compounding errors, while DADP-I maintains low deviation throughout execution. These findings confirm that inverse dynamics guidance effectively suppresses error propagation and improves physical consistency in long-horizon planning.

Table 1: Normalized scores on D4RL Maze2D and Multi2D benchmarks.

| Environment | BCQ | CQL | IQL | ROMI | DD | HDMI | Diffuser | Diffuser (HER) | Diffuser (AM) | DPCC | DADP-F (Ours) | DADP-I (Ours) |
|---|---|---|---|---|---|---|---|---|---|---|---|---|
| Maze2D U-Maze | 12.8 | 5.7 | 47.4 | **139.5** | 116.2 | 120.1 | 113.9 | 125.4 | 121.4 | 50.2 | 130.2 ± 0.8 | 131.5 ± 0.7 |
| Maze2D Medium | 8.3 | 5.0 | 34.9 | 82.4 | 122.3 | 121.8 | 121.5 | 130.3 | 127.2 | 63.9 | **135.4** ± 0.5 | **136.8** ± 0.8 |
| Maze2D Large | 6.2 | 12.5 | 58.6 | 83.1 | 125.9 | 128.6 | 123.0 | 135.8 | 135.0 | 96.5 | **147.7** ± 3.6 | 144.2 ± 4.1 |
| **Single-task Average** | 9.1 | 7.7 | 47.0 | 101.7 | 121.5 | 123.5 | 119.5 | 130.5 | 127.9 | 70.2 | **137.8** | 137.5 |
| Multi2D U-Maze | - | - | 24.8 | - | 128.2 | 131.3 | 128.9 | 132.7 | 135.4 | 86.4 | **136.8** ± 1.3 | 137.0 ± 1.6 |
| Multi2D Medium | - | - | 12.1 | - | 129.7 | 131.6 | 127.2 | 133.0 | 137.8 | 76.8 | **139.7** ± 0.7 | 142.1 ± 0.9 |
| Multi2D Large | - | - | 13.9 | - | 130.5 | 135.4 | 132.1 | 139.2 | 145.7 | 112.7 | **156.1** ± 2.2 | 158.7 ± 2.5 |
| **Multi-task Average** | - | - | 16.9 | - | 129.5 | 132.8 | 129.4 | 135.0 | 139.6 | 91.9 | **144.2** | 145.9 |

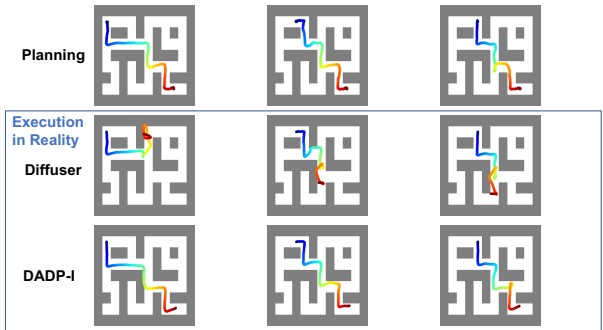

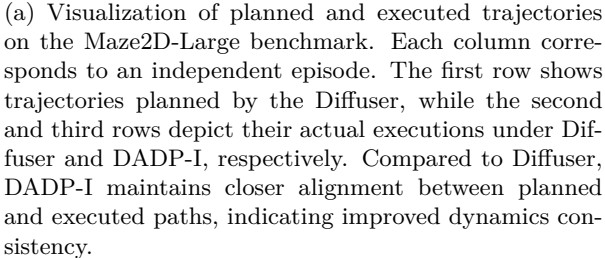

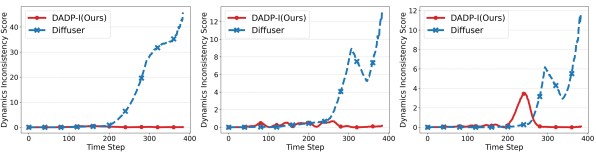

(a) Visualization of planned and executed trajectories on the Maze2D-Large benchmark. Each column corresponds to an independent episode. The first row shows trajectories planned by the Diffuser, while the second and third rows depict their actual executions under Diffuser and DADP-I, respectively. Compared to Diffuser, DADP-I maintains closer alignment between planned and executed paths, indicating improved dynamics consistency.

(b) Dynamics inconsistency scores for the three episodes illustrated in (a). Each plot shows the deviation between planned and executed trajectories for Diffuser and DADP-I. Lower scores indicate higher fidelity and better execution consistency.

Figure 3: Comparison between trajectory visualization and dynamics consistency analysis on the Maze2D-Large benchmark. (a) visualizes planned and executed trajectories, while (b) reports corresponding dynamics inconsistency scores (measured as the Euclidean distance against the ground-truth simulator states as defined in Eq. 9). Lower scores indicate higher fidelity and better execution consistency.

## 4.2 MuJoCo: Robust Continuous Control

**Datasets.** We further evaluate DADP on the D4RL MuJoCo continuous-control benchmarks, including HalfCheetah, Hopper, and Walker2d. Each environment is evaluated under three datasetsmedium-expert, medium, and medium-replaythat capture increasing degrees of data imperfection and distributional shift. Each dataset contains one million transitions generated by policies of different quality levels. The medium-expert dataset combines trajectories from expert and medium policies in equal proportions, the medium dataset is derived from a partially trained Soft ActorCritic (SAC) policy, and the medium-replay dataset aggregates all replay buffer transitions up to a medium-performance policy.

**Baselines.** We compare DADP with representative offline reinforcement learning algorithms from both model-free and model-based paradigms. Model-free baselines include Behavior Cloning (BC), Conservative Q-Learning (CQL) (Kumar et al., 2020), Implicit Q-Learning (IQL) (Kostrikov et al., 2021), and Decision Transformer (DT) (Chen et al., 2021). For model-based approaches, we consider MOPO (Yu et al., 2020) and Diffuser (Janner et al., 2022). This comprehensive set of baselines enables a fair comparison across policy-learning, model-based, and diffusion-based paradigms, isolating the contribution of dynamics alignment in improving robustness and stability.

**Results.** Table 2 reports normalized scores across all MuJoCo environments and datasets. Both DADP-F and DADP-I show superior performance compared to most baseline methods, where DADP-I attains the highest average normalized score of 81.2 across all evaluated tasks. DADP-F excels on tasks with smoother and more predictable dynamics, such as HalfCheetah and Walker2d, whereas DADP-I demonstrates superior robustness in stochastic environments like Hopper. Compared to Diffuser, both variants achieve substantial gainsparticularly on the medium and medium-replay datasetsindicating that explicit dynamics alignment effectively mitigates model bias and stabilizes long-horizon control. Overall, these results confirm that incorporating forward or inverse dynamics guidance significantly enhances the stability, robustness, and generalization of diffusion-based offline reinforcement learning.

Table 2: Performance comparison on the D4RL MuJoCo benchmark. Each entry reports normalized scores for different offline RL algorithms across three datasets (medium-expert, medium, and medium-replay). Results for DADP-F and DADP-I are averaged over **100 evaluation episodes** and reported as mean ś standard error. The highest score for each task is highlighted in bold.

| Dataset | Environment | CQL | IQL | DT | MOPO | Diffuser | DADP-F (Ours) | DADP-I (Ours) |
|---------|-------------|-----|-----|-----|------|----------|---------------|---------------|
| Medium-Expert | HalfCheetah | 91.6 | 86.7 | 86.8 | 63.3 | 88.9 | 89.6 ± 0.1 | **92.3** ± 0.1 |
| Medium-Expert | Hopper | 105.4 | 91.5 | 107.6 | 23.7 | 103.3 | 104.4 ± 1.0 | **108.1** ± 1.0 |
| Medium-Expert | Walker2d | 108.8 | **109.6** | 108.1 | 44.6 | 106.9 | 107.6 ± 0.1 | 108.0 ± 0.1 |
| Medium | HalfCheetah | 44.0 | **47.4** | 42.6 | 42.3 | 42.8 | 44.9 ± 0.1 | 43.9 ± 0.1 |
| Medium | Hopper | 58.5 | 66.3 | 67.6 | 28.0 | 74.3 | 77.3 ± 0.7 | **96.2** ± 0.7 |
| Medium | Walker2d | 72.5 | 78.3 | 74.0 | 17.8 | 79.6 | **81.8** ± 0.4 | 80.5 ± 0.4 |
| Medium-Replay | HalfCheetah | 45.5 | 44.2 | 36.6 | **53.1** | 37.7 | 38.7 ± 0.3 | 38.8 ± 0.3 |
| Medium-Replay | Hopper | 95.0 | 94.7 | 82.7 | 67.5 | 93.6 | 95.2 ± 0.4 | **95.6** ± 0.5 |
| Medium-Replay | Walker2d | 77.2 | 73.9 | 66.6 | 39.0 | 70.6 | **77.9** ± 2.4 | 71.9 ± 1.8 |
| **Average** | | 77.6 | 77.0 | 74.7 | 42.1 | 77.5 | **79.7** | **81.2** |

## 4.3 Ablation Study and Sensitivity Analysis

To assess the contribution of the dynamics-alignment mechanism, we conduct ablation experiments on both the Maze2D and MuJoCo benchmarks by removing the dynamics-consistency term from DADP. As shown in Figure 4, excluding this term (DADP w/o Alignment) leads to a consistent drop in normalized scores across all Maze2D variants under both single-task and multi-task settings. This confirms that the alignment component effectively mitigates long-horizon error accumulation and enforces physical feasibility during trajectory generation.

We further extend the ablation to the MuJoCo control tasks (Table 3), where both DADP-F and DADP-I out-perform the unaligned variant across all datasets. DADP-F performs better in environments with smoother

dynamics such as HalfCheetah and Walker2d, while DADP-I exhibits higher robustness in stochastic and underactuated environments such as Hopper. These results demonstrate that dynamics alignment provides consistent benefits across diverse control regimes and varying environment complexities.

To analyze the robustness of DADP with respect to hyperparameter choices, we perform sensitivity studies on the inverse-aligned variant (DADP-I) using the MuJoCo benchmark. Here, $\lambda$ controls the weighting of the reward guidance term, and $\beta$ adjusts the strength of the dynamics-alignment constraint during denoising. The step-size coefficient $\alpha$ is found to be relatively insensitive and is fixed to 1 in all experiments. Figure 5 presents the sensitivity curves for $\lambda$ (top) and $\beta$ (bottom) across three environments: HalfCheetah, Hopper, and Walker2d. Performance remains stable over several orders of magnitude for both parameters, highlighting that DADP-I is robust to hyperparameter variations and requires minimal tuning effort.

**Discussion on Combining Forward and Inverse Guidance.** A natural question arises: can we combine DADP-F and DADP-I to enforce both state and action consistency simultaneously? While theoretically appealing, this combination introduces practical challenges. First, calculating gradients from two distinct dynamics models doubles the computational overhead during inference. Second, jointly optimizing for reward, forward consistency, and inverse consistency requires balancing three weighting hyperparameters ($\lambda, \beta_F, \beta_I$), which can lead to gradient conflicts and increased tuning complexity. Empirically, we found that selecting the variant best suited for the task structure (e.g., DADP-I for complex control, DADP-F for navigation) yields significant gains while maintaining algorithmic simplicity. We leave the efficient unification of these objectives to future work.

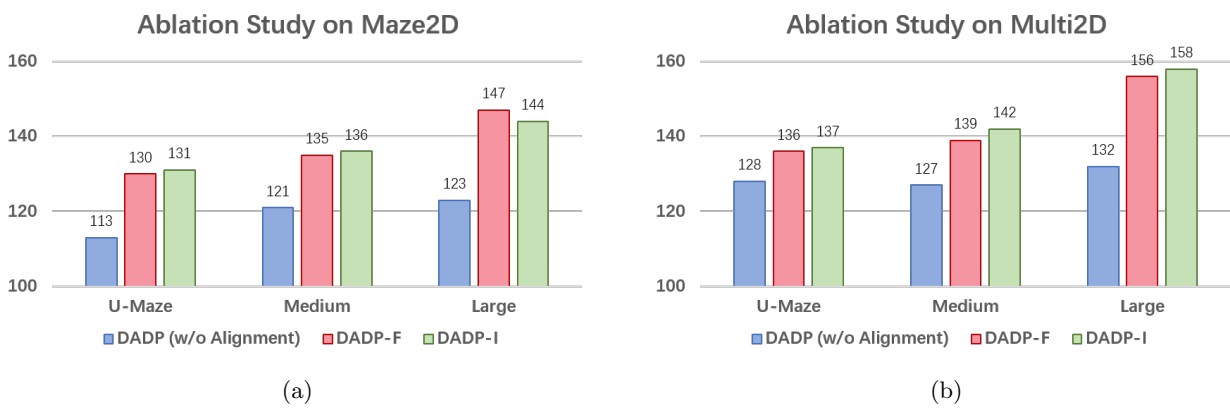

Figure 4: Ablation study results on the Maze2D and Multi2D benchmarks. (a) Maze2D represents single-task planning with fixed start and goal locations, while (b) Multi2D evaluates the multi-task setting where both start and goal positions are randomly sampled across episodes. Each bar shows the normalized score of DADP variants with and without dynamics alignment.

Table 3: Ablation study results on the D4RL MuJoCo benchmark. Each dataset group (Medium-Expert, Medium, Medium-Replay) contains three continuous control environments: HalfCheetah, Hopper, and Walker2d.

| Method | Medium-Expert | | | Medium | | | Medium-Replay | | |
|---|---|---|---|---|---|---|---|---|---|
| | HalfCheetah | Hopper | Walker2d | HalfCheetah | Hopper | Walker2d | HalfCheetah | Hopper | Walker2d |
| DADP (w/o Alignment) | 88.9 | 103.3 | 106.9 | 42.8 | 74.3 | 79.6 | 37.7 | 93.6 | 70.6 |
| DADP-F | 89.6 | 104.4 | 107.6 | 44.9 | 77.3 | 81.8 | 38.7 | 95.2 | 77.9 |
| DADP-I | 92.3 | 108.1 | 108.0 | 43.9 | 96.2 | 80.5 | 38.8 | 95.6 | 71.9 |

## 4.4 Open-Loop Multi-Step Execution Analysis

Most previous diffusion-based planners adopt a closed-loop execution scheme, where the agent replans after every action using the most recent observation. While this strategy effectively limits error accumulation, it

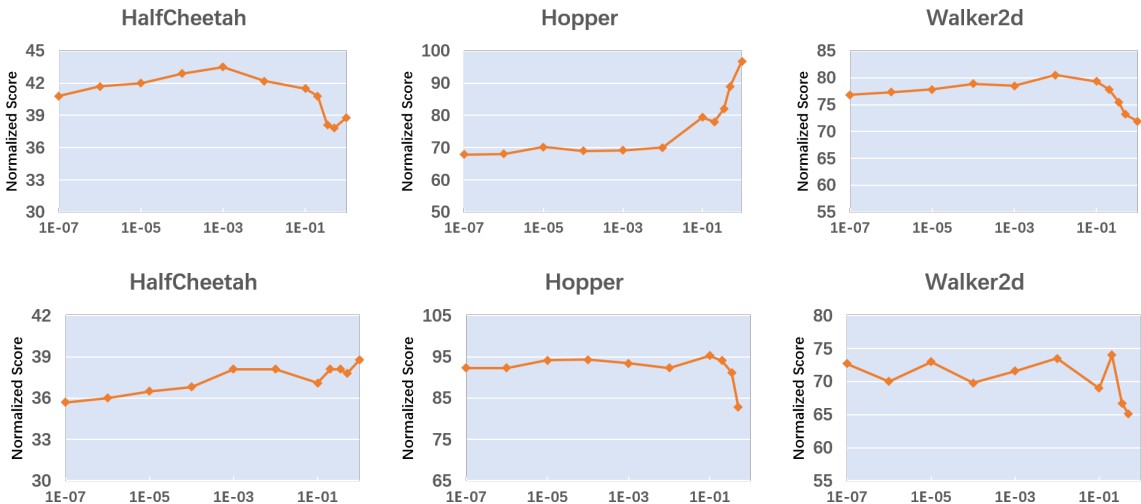

Figure 5: Sensitivity analysis of hyperparameters $\lambda$ and $\beta$ on the MuJoCo benchmark. The top figure reports results on the Medium setting, showing the effect of $\lambda$, which controls the weighting of the reward objective. The bottom figure shows the Medium-Replay setting, illustrating the influence of $\beta$, the dynamics alignment weight. Both $\lambda$ and $\beta$ are plotted on a logarithmic scale.

incurs substantial computational overhead. In practical scenarios, however, reducing planning frequency by executing multiple actions per planan open-loop strategycan greatly improve efficiency.

Figure 6 illustrates the trade-off between average rollout time and performance for DADP-I on the MuJoCo Hopper-medium task. As the number of executed actions per plan increases, the overall computation time decreases substantially, with only marginal degradation in normalized performance.

These results demonstrate that DADP-I maintains strong robustness under reduced planning frequency, achieving an effective balance between computational efficiency and control performancean essential property for real-time deployment of diffusion-based planners.

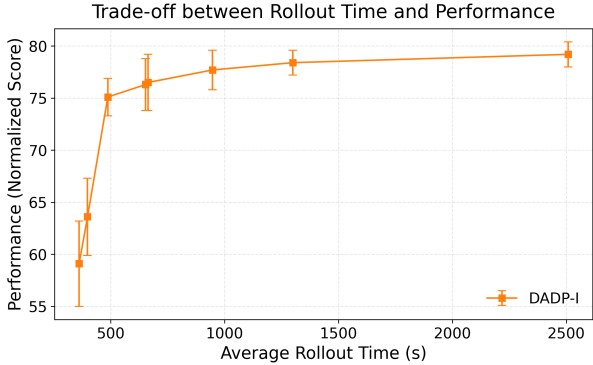

Figure 6: Trade-off between rollout time and performance on the MuJoCo Hopper-medium task. Each point represents the average performance of DADP-I under different open-loop execution lengths. The results show that DADP-I maintains stable performance as rollout time decreases, achieving an effective balance between computational efficiency and control accuracy.

# 5 Limitations and Future Work

Despite its promising results, DADP faces several limitations. First, the iterative inference of diffusion models incurs high computational costs, limiting scalability to real-time applications. Second, while we employ probabilistic VAEs to capture dynamics, our current guidance mechanism relies on the predicted mean with a fixed weight ($\gamma_t = 1$). This formulation does not fully exploit the model's uncertainty estimates, potentially introducing model bias under severe distribution shifts. Consequently, the planning performance is effectively upper-bounded by the fidelity of the learned dynamics. Incorporating adaptive weighting based on VAE variance represents a promising avenue to further enhance safety in uncertain regions. Finally, DADP assumes static offline datasets without online adaptation. Furthermore, our current evaluation is limited to fully observable, state-based continuous control tasks. Applying DADP to high-dimensional pixel-based observations, partially observable MDPs, or contact-rich manipulation tasks would likely require integrating latent dynamics models or world models, which we identify as a key area for future work.

# 6 Conclusion

This paper presented Dynamics-Aligned Diffusion Planning (DADP), a unified framework that integrates explicit dynamics modeling into diffusion-based trajectory generation to address dynamics inconsistency in offline reinforcement learning. By incorporating forward and inverse dynamics guidance, DADP enforces physical feasibility during denoising and enhances stability in long-horizon planning. Extensive experiments on the D4RL Maze2D and MuJoCo benchmarks demonstrated that DADP significantly improved planning accuracy, robustness, and generalization over prior diffusion-based methods.

# Broader Impact Statement

This work introduces Dynamics-Aligned Diffusion Planning (DADP) to improve the reliability of offline reinforcement learning in complex control tasks. While our method demonstrates enhanced planning consistency, we acknowledge potential risks and societal impacts associated with its deployment in real-world scenarios.

**Safety in Critical Applications.** Our paper motivates the use of DADP in domains such as autonomous driving and medical treatment. However, the framework relies on *learned* dynamics models (VAEs) to enforce consistency. In safety-critical settings, learned models may suffer from hallucinations or prediction errors, particularly when the agent encounters out-of-distribution (OOD) states not well-represented in the offline dataset. Relying solely on these learned priors without rigorous uncertainty quantification or hard safety constraints (e.g., rule-based shields) could lead to dangerous failures. Practitioners should exercise caution and consider integrating DADP with verifiable safety protocols before real-world deployment.

**Computational Cost and Environmental Impact.** Diffusion-based planning involves iterative denoising processes, which incur significantly higher computational costs and inference latency compared to standard MLP-based policies (e.g., CQL, IQL). This increased computational burden translates to higher energy consumption and a larger carbon footprint, especially during large-scale training and deployment. Future research should focus on improving the sampling efficiency of DADP (e.g., via distillation or fast samplers) to mitigate its environmental impact and enable deployment on resource-constrained edge devices.

### Acknowledgments

This work is partially supported by National Natural Science Foundation of China (62476128).

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

# A    Dataset Details

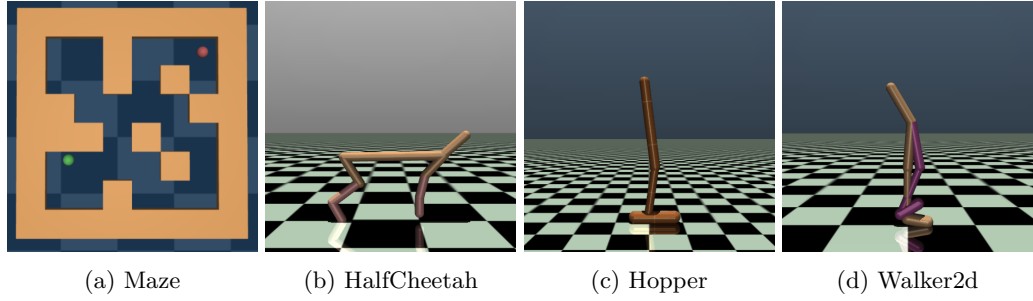

|        (a) Maze        |      (b) HalfCheetah      |       (c) Hopper       |      (d) Walker2d      |

Figure 7: (a) Maze: A point-mass (green dot) agent needs to plan a path from the starting position to reach the target location (red dot). (b) HalfCheetah: A two-dimensional cheetah robot that controls its joints to run as fast as possible to the right. (c) Hopper: A two-dimensional monopod robot where the agent needs to control its hinge to perform a rightward hopping motion. (d) Walker2d: Based on Hopper, an additional leg is added, requiring the agent to coordinate the movements of both legs to achieve fast walking while maintaining balance.

## A.1    Maze2D

The Maze2D benchmark in D4RL (Fu et al., 2020) is designed for evaluating long-horizon navigation and sparse-reward planning. It consists of three environments: U-Maze, Medium, and Large, which progressively increase in complexity and trajectory length. Each dataset contains trajectories collected by a goal-conditioned controller navigating through 2D mazes of varying sizes. The agents state consists of the (x, y) location and velocities. A sparse reward of +1 is provided only upon reaching the goal, while intermediate

steps receive zero reward. Following the standard D4RL evaluation protocol, we compute normalized scores by first normalizing the cumulative episode rewards between the minimum and maximum returns of reference (random and expert) policies, and then scaling the result by 100. Thus, a score of 100 corresponds to the expert-level performance, while 0 represents random behavior. The final score is obtained by averaging normalized returns over all evaluation episodes.

## A.2  MuJoCo

The MuJoCo benchmark in D4RL contains three standard continuous control tasks: HalfCheetah, Hopper, and Walker2d. Each task is evaluated on three offline datasetsmedium-expert, medium, and medium-replayeach containing one million transitions generated by Soft Actor-Critic (SAC) agents trained to different degrees of proficiency.

- **Medium-Expert:** 50% expert and 50% medium-level trajectories mixed uniformly.

- **Medium:** trajectories from a partially trained SAC policy paused at mid-training.

- **Medium-Replay:** all transitions from the replay buffer during SAC training up to the medium performance level.

All environments provide continuous state and action spaces with dense rewards defined by task-specific physical performance metrics (e.g., forward velocity, energy cost, stability). We follow the D4RL normalization protocol, where a score of 0 corresponds to a random policy and 100 to an expert policy.

## A.3  Data Usage Protocol

All experiments strictly follow the experimental setup of Diffuser (Janner et al., 2022). Datasets are used exactly as provided by D4RL without any additional filtering, augmentation, or relabeling. We ensure fair comparison by adopting identical training and evaluation splits across all methods. To ensure statistical reliability, each experiment is performed using **3 independent training seeds**. Unlike deterministic policies, our diffusion-based planner is stochastic, requiring extensive evaluation rollouts. Consequently, the reported results correspond to the aggregate of **150 evaluation episodes** for Maze2D (3 seeds × 50 rollouts) and **100 evaluation episodes** for MuJoCo, strictly following the standard evaluation protocols of the baselines. All reported results include the mean and standard error across the training seeds.

# B  Model and Network Architecture Details

## B.1  Diffusion Model

The diffusion model used in DADP follows the same architecture as the original Diffuser (Janner et al., 2022). It adopts a one-dimensional temporal U-Net composed of six repeated residual blocks, designed to capture hierarchical temporal dependencies within trajectory sequences. Each residual block contains two temporal convolution layers, each followed by group normalization (Wu & He, 2018) and a Mish activation function (Misra, 2019). Timestep embeddings are generated by a single fully connected layer and are added to the activation of the first temporal convolution within each block to provide diffusion-step awareness. This structure enables stable denoising in the trajectory space while maintaining temporal smoothness and inter-step consistency.

## B.2  Return Predictor Architecture

The return predictor $J(\tau)$ utilizes the same **temporal U-Net encoder backbone** as the diffusion model. This shared architecture ensures feature compatibility and representational consistency. Structurally, the return predictor retains the encoder layers but replaces the decoder with a final Multi-Layer Perceptron (MLP) projection head. This head maps the aggregated trajectory embedding to a scalar value representing the predicted cumulative return.

### B.3 Dynamics Model

The dynamics model, used for both forward and inverse consistency guidance, is implemented as a variational autoencoder (VAE). Each VAE consists of a three-layer fully connected encoder and decoder, each with a hidden dimension of 256 and ReLU activations. The encoder outputs the mean and log standard deviation of a 32-dimensional latent variable $z$, sampled via the reparameterization trick. The latent variable captures stochasticity and multi-modality in the environment dynamics, enabling smooth and differentiable consistency gradients.

## C Training Configuration

### C.1 Optimization and Training Setup

All models are trained using the Adam optimizer with default momentum parameters. For the diffusion model, we use a learning rate of $2 \times 10^{-4}$ and a batch size of 32. For the dynamics models (both forward and inverse), the learning rate is set to $1 \times 10^{-3}$ with a batch size of 64.

All models are trained using the same datasets as those used for the diffusion process. Each task (Maze2D or MuJoCo) is trained on its corresponding D4RL dataset independently, ensuring that the diffusion and dynamics models are both exposed to consistent trajectory distributions. This alignment between training datasets guarantees that the dynamics priors provide coherent and task-specific consistency guidance during planning.

### C.2 Diffusion and Return Predictor Training

The diffusion model is trained following the standard denoising diffusion probabilistic modeling (DDPM) objective:

$$\mathcal{L}_{\text{diff}} = \mathbb{E}_{\tau,\epsilon,i}\big[\|\epsilon - \epsilon_\theta(\tau^i, i)\|_2^2\big], \tag{21}$$

where $\epsilon_\theta$ denotes the noise prediction network parameterized by the temporal U-Net. We set the diffusion horizon $N = 20$ in MuJoCo and $N = 256$ in Maze2D.

The return predictor is trained via supervised regression to minimize the Mean Squared Error (MSE) between the predicted and actual discounted cumulative returns $\mathcal{J}(\tau) = \sum_{t=0}^{T-1} \gamma^t r(s_t, a_t)$. To ensure training stability and consistent gradient scaling, we apply a **GaussianNormalizer** to standardize the target returns (zero mean, unit variance) based on statistics computed from the training dataset. We use the exact same training and validation splits as the diffusion model to prevent data leakage.

### C.3 Dynamics Model Training

Both the forward and inverse dynamics models are trained independently using the variational autoencoder (VAE) objective. Each model employs a latent dimension of 32, with both encoder and decoder composed of three fully connected layers of size 256, each followed by ReLU activation. The encoder outputs the mean and log standard deviation of the latent variable, which is sampled via the reparameterization trick for differentiability.

Training is conducted using the Adam optimizer with a learning rate of $1 \times 10^{-3}$ and a batch size of 64. Each model is trained for 3K epochs using the same dataset as the diffusion model, ensuring consistent trajectory distributions across components. Model checkpoints are saved periodically (every 1000 epochs) and the best model is selected based on validation loss. All training runs employ the same data preprocessing and normalization pipeline as in the diffusion model, with no data augmentation or relabeling applied.

### C.4 Hyperparameter Settings

During planning, the key guidance coefficients are selected per environment via validation: $\alpha = 1.0$ (step size), $\lambda \in [0.8, 1.0]$ (reward weight), and $\beta \in [10^{-2}, 10^0]$ (dynamics alignment weight). These ranges balance reward optimization and dynamics consistency.

### C.5 Hardware and Software Configurations

All experiments are conducted on a workstation equipped with two Intel Xeon Gold 5218 CPUs (2.30GHz, 32 cores in total), four NVIDIA GeForce RTX 2080 Ti GPUs (11GB each), and 256GB of system memory. The operating system is Ubuntu Linux with CUDA 13.0 and driver version 580.95.05.

## D  Dynamics Model Accuracy

To ensure the reliability of the dynamics guidance, we evaluated the prediction accuracy of both the forward and inverse dynamics models on a held-out test set (10% of the dataset). Table 4 reports the Mean Squared Error (MSE) for state prediction (Forward) and action reconstruction (Inverse). Note that the absolute magnitude of MSE varies across environments due to differences in state dimensionality and feature scales (e.g., HalfCheetah involves higher-velocity state components compared to Maze2D). Overall, the consistent low test errors relative to the scale of each environment confirm that the learned models capture the underlying dynamics effectively and provide accurate gradients for trajectory refinement.

Table 4: Held-out Test Mean Squared Error (MSE) for the learned Forward and Inverse dynamics models. The results demonstrate that the models achieve reasonable prediction accuracy across diverse datasets, ensuring reliable gradient guidance for the diffusion planner.

| Environment | Forward Model Test MSE | Inverse Model Test MSE |
|---|---|---|
| *MuJoCo Benchmarks* | | |
| HalfCheetah Medium-Expert | 0.8538 | 0.0086 |
| HalfCheetah Medium | 0.9465 | 0.0089 |
| HalfCheetah Medium-Replay | 1.1733 | 0.0148 |
| Hopper Medium-Expert | 0.0360 | 0.0320 |
| Hopper Medium | 0.0373 | 0.0254 |
| Hopper Medium-Replay | 0.0762 | 0.0486 |
| Walker2d Medium-Expert | 0.3338 | 0.0322 |
| Walker2d Medium | 0.3498 | 0.0320 |
| Walker2d Medium-Replay | 0.8267 | 0.0879 |
| *Maze2D Benchmarks* | | |
| Maze2D U-Maze | 0.0014 | 0.0183 |
| Maze2D Medium | 0.0022 | 0.0150 |
| Maze2D Large | 0.0016 | 0.0204 |

## E  Efficiency and Latency Analysis

### E.1 Sampling Steps vs. Execution Horizon

**Sampling Steps ($N$) Constraints.** A potential strategy to accelerate inference is reducing the number of diffusion denoising steps $N$. However, in our implementation, the diffusion model is trained with a fixed noise schedule corresponding to $N$ steps (e.g., $N = 20$ for MuJoCo). Unlike consistency models or DDIM, directly reducing $N$ during inference without retraining typically leads to a misalignment between the reverse process and the learned dynamics, resulting in dynamically infeasible trajectories. Since our current $N$ is already relatively compact to ensure fast planning, further reducing it yields diminishing returns in latency while severely compromising performance.

**Execution Horizon Trade-off.** Instead, we identify **Open-Loop Multi-Step Execution** (as analyzed in Section 4.4) as a more effective lever for optimizing the efficiency-accuracy trade-off. By executing a sequence of $k$ actions from a single planned trajectory before replanning, we amortize the inference cost of the diffusion model over $k$ steps. As shown in Figure 6, this strategy significantly reduces the effective latency per step with minimal impact on control performance, providing a practical solution for real-time deployment.

### E.2 Wall-Clock Inference Latency

To explicitly quantify the computational overhead introduced by the dynamics alignment process, we measure the average wall-clock time required to generate a single plan (batch size = 1) on an NVIDIA GeForce RTX 2080 Ti GPU. Table 5 details the inference latency across three MuJoCo environments. Since DADP requires backpropagating gradients through the dynamics VAE at each denoising step, it incurs a slight additional computational cost. However, as shown in the table, the relative slowdown is consistent and marginal ($\approx 1.05\times$) across different tasks. Combined with the open-loop execution strategy discussed above, this minimal overhead ensures that DADP remains highly efficient for practical deployment.

Table 5: Comparison of average wall-clock inference latency per plan (in seconds). Results are averaged over 100 independent planning steps on a single NVIDIA GeForce RTX 2080 Ti GPU. The results demonstrate that DADP introduces consistent and minimal latency overhead ($\sim 5\%$) compared to the baseline Diffuser.

| Environment | Diffuser (Baseline) | DADP-F (Ours) | DADP-I (Ours) |
|---|---|---|---|
| HalfCheetah-Medium-Replay | $1.16 \pm 0.12$ | $1.22 \pm 0.03$ | $1.22 \pm 0.08$ |
| Hopper-Medium-Replay | $1.81 \pm 0.03$ | $1.92 \pm 0.10$ | $1.88 \pm 0.13$ |
| Walker2d-Medium-Replay | $1.77 \pm 0.12$ | $1.80 \pm 0.18$ | $1.87 \pm 0.14$ |
| **Average Slowdown** | **1.00x** | **1.04x** | **1.05x** |

