# OpenReview forum: "Dynamics‑Aligned Diffusion Planning for Offline RL: A Unified Framework with Forward and Inverse Guidance"
_TMLR — Accepted by TMLR_

### Review · Reviewer_rjDr · 2025-12-03

**Summary Of Contributions:**

# Summary

The paper proposes Dynamics-Aligned Diffusion Planning (DADP), a trajectory diffusion framework for offline RL that explicitly enforces consistency with environment dynamics during reverse denoising. Two instantiations are presented: DADP-F uses a learned forward model to penalize state-level transition inconsistency, while DADP-I uses a learned inverse model to penalize action-level infeasibility; both are integrated via a unified guidance term that balances reward and dynamics-alignment gradients. Experiments on D4RL Maze2D and MuJoCo show that both variants outperform diffusion-based and standard offline RL baselines, with ablations and a planning-frequency study indicating improved long-horizon robustness and computational trade-offs.

# Strengths

- The paper addresses a concrete and recurring failure mode in diffusion planning, namely denoised trajectories that look globally coherent but violate local dynamics, hence leading to compounding error over horizons.
- The approach is simple to implement on top of existing denoisers and thus potentially impactful for the growing body of diffusion-based decision-making.
- The paper introduces a unified, energy-guided trajectory denoising scheme that incorporates differentiable dynamics consistency, with complementary forward (state-level) and inverse (action-level) variants.
- The paper employs VAE-based forward/inverse models to provide smooth, stochastic guidance signals; this is pragmatic and plausibly more robust than purely deterministic dynamics regressors.
- A Clean guidance formulation $g = \nabla \mu (\lambda J − \beta E)$ that directly augments the reverse diffusion mean, compatible with existing Diffuser-style planners.
- The paper presents a broad evaluation on D4RL Maze2D and MuJoCo across multiple datasets, with consistent improvements over Diffuser and strong offline RL baselines. Useful ablations (w/o alignment) and sensitivity studies for $\lambda$ and $\beta$; open-loop multi-step execution analysis addresses a practical compute-performance trade-off.
- A qualitative and quantitative analysis of “dynamics inconsistency” (planned vs executed trajectory divergence) strengthens the motivation.

# Weaknesses

- The claimed “theoretical insight” that alignment mitigates compounding error is not substantiated with formal results or nontrivial analysis in the presented text.
- Guidance weights do not explicitly incorporate model uncertainty (despite using VAEs), and $\gamma_t$ is fixed to 1 in experiments; ignoring variance calibration may cause mis-weighted gradients in regions with high model uncertainty.
- Reliance on learned dynamics models introduces another layer of model bias; the framework does not quantify or adaptively mitigate this bias beyond fixed $\lambda$, $\beta$ schedules.
- Missing comparisons to closely related constrained or dynamics-aware diffusion approaches that also inject dynamics into reverse sampling, such as DPCC (projection with dynamics), Equality-Constrained Diffusion (augmented Lagrangian for dynamics constraints), and Constrained Diffusers (primal-dual/ALM constrained denoising with an inverse dynamics layer). These are natural baselines for the core claim of “dynamics alignment.”
- No analysis of dynamics model calibration or accuracy (e.g., negative log-likelihoods, calibration curves, or how guidance reacts to mis-specified models). This is critical to substantiate the claim that alignment reduces compounding error rather than simply steering toward another biased manifold.
- The paper proposes two standalone variants (forward vs inverse) but does not evaluate their combination, which seems straightforward under the unified guidance and could be stronger.
- Details about the return predictor training (data splits, targets, normalization, potential leakage) are not fully specified; in MuJoCo, whether reward is computed analytically or via a learned model is ambiguous.
- The “dynamics inconsistency” metric is not fully operationalized beyond $L_2$ distances; whether this is measured against the true simulator or learned models differs across sections, and the choice materially affects conclusions.

**Audience:**

Yes

**Audience Explanation:**

The paper addresses a timely and practically relevant problem at the intersection of two active research areas: diffusion models for decision-making and offline reinforcement learning. The phenomenon of "dynamics inconsistency", where diffusion-generated trajectories appear globally coherent but violate local transition constraints, is a recognized failure mode that practitioners encounter. The proposed solution of integrating forward/inverse dynamics guidance into the denoising process is conceptually simple, easy to implement atop existing diffusion planners, and broadly applicable. Researchers working on diffusion-based control, model-based RL, robotics planning, and offline RL would likely find the framework and empirical findings relevant to their work.

**Broader Impact Concerns:**

Since the paper motivates the work with safety-critical applications like medical treatment and autonomous driving, a Broader Impact Statement should explicitly address the risks of relying on learned dynamics models for safety constraints, where model errors could lead to dangerous failures. Additionally, the significant computational cost of diffusion-based planning  warrants a discussion on energy consumption and the environmental impact of large-scale deployment.

**Claims And Evidence:**

No

**Claims Explanation:**

While the paper presents substantial empirical results, several claims lack adequate support:
- **Unsupported theoretical claim:** The paper lists "Theoretical Insight: We provide an analysis showing that explicit dynamics alignment mitigates compounding errors" as a main contribution. However, no formal analysis, proofs, or non-trivial theoretical results are provided—only empirical observations (Figure 3b). This overclaims the contribution.
- **Missing critical baselines:** The core claim of "dynamics alignment" is not compared against closely related constrained diffusion methods (DPCC, Equality-Constrained Diffusion, Constrained Diffusers) that also inject dynamics into reverse sampling. Without these comparisons, the claimed novelty and superiority cannot be fully evaluated.
- **Methodological inconsistency regarding uncertainty:** The authors motivate the use of VAE-based dynamics models specifically to "capture uncertainty" and enable "uncertainty-aware consistency guidance"1. However, the experimental setup contradicts this by using a fixed uniform weighting ($\gamma_t=1$). This effectively discards the uncertainty information the VAE provides, rendering the claim of "probabilistic" or "uncertainty-aware" guidance unsupported by the actual implementation.
- **Unverified experimental details:** The reported 100-150 random seeds is unusually high for D4RL evaluations (typical is 3-10 seeds). This requires clarification, as it affects the credibility of the statistical claims.
- **Missing model validation:** No calibration or accuracy metrics are reported for the learned dynamics models, yet the entire framework depends on their fidelity. This gap undermines confidence that the method reduces compounding error rather than introducing different biases.

**Requested Changes:**

Refer to the weaknesses above.

---

> ### Author Response · Authors · 2025-12-22
> **Response to Reviewer rjDr (Part 1/3)**
>
> **Dear Reviewer rjDr,**
>
> Thank you for the constructive feedback and for recognizing the effectiveness of DADP in addressing dynamics inconsistency. We have carefully revised the manuscript to ensure precise terminology and clearer positioning of our contributions. Below is our point-by-point response.
>
> ---
>
> ### 1. Clarification of "Theoretical Insight"
>
> **Reviewer Comment:** *The claimed “theoretical insight” is not substantiated with formal results... This overclaims the contribution.*
>
> **Response:**
>
> We appreciate the emphasis on terminological rigor.
>
> We would like to clarify that our use of the term "theoretical insight" was intended to highlight the **mechanistic understanding** derived from our analysis—specifically, identifying *how* dynamics inconsistency leads to compounding errors in diffusion planning. This analysis (Fig. 3b) goes beyond performance metrics to explain the underlying cause of trajectory divergence, which constitutes a conceptual contribution.
>
> However, we agree with the Reviewer that in a strict mathematical context, "theoretical" often implies the presence of formal lemmas or error bounds. To avoid any potential ambiguity or misalignment with formal expectations, we are happy to adopt more precise terminology that accurately reflects the nature of our findings without diminishing their analytical value.
>
> **Action:**
>
> We have revised the manuscript to characterize this contribution more precisely:
>
> 1. **Refined Terminology:** In the **Introduction** and relevant sections, we have updated the phrasing from "Theoretical Insight" to **"Principled Formulation and Mechanistic Analysis."**
> 2. **Clarified Narrative:** We now explicitly state that our contribution lies in **deriving** the exact guidance gradient (Eq. 16) from an optimization-as-inference perspective. This highlights that our findings are grounded in a rigorous mathematical formulation, distinct from heuristic engineering, without overclaiming the existence of convergence proofs.
>
> ### 2. Comparison to Related Constrained Diffusion Methods
>
> **Reviewer Comment:** *Missing comparisons to closely related constrained or dynamics-aware diffusion approaches (e.g., DPCC, Equality-Constrained Diffusion).*
>
> **Response:**
>
> We thank the Reviewer for highlighting these relevant works. We agree that they provide important context regarding dynamics integration.
>
> While we share the high-level motivation of ensuring physical plausibility, a direct quantitative comparison (e.g., on D4RL benchmarks) is not methodologically suitable due to fundamental divergences in **Problem Formulation**, **Constraint Philosophy**, and **Inference Efficiency**. We detail these distinctions below:
>
> 1.  **Distinct Problem Formulations (Stitching vs. Imitation/Control):**
>     *   **DADP (Ours):** Focuses on **Offline RL**, where the core challenge is "trajectory stitching"—aggregating suboptimal segments to form a super-optimal policy. DADP uses dynamics primarily to prevent value overestimation during this stitching process.
>     *   **Baselines (DPCC, etc.):** These methods generally target **Imitation Learning** (reproducing expert behavior) or **Optimal Control** (solving for feasibility). They are not designed to maximize rewards beyond the dataset quality. Comparing them on Offline RL metrics would be inequivalent, as their objectives are not aligned with policy improvement from suboptimal data.
>
> 2.  **Robustness to Model Uncertainty (Soft vs. Hard Constraints):**
>     *   Approaches like *Equality-Constrained Diffusion* treat dynamics as **hard constraints**. This is effective when the dynamics model is analytical or perfect.
>     *   In Offline RL, the dynamics model is *learned* and inherently contains error (aleatoric/epistemic uncertainty). Enforcing hard constraints on an imperfect model can force the planner into adversarial regions or cause convergence failures. DADP deliberately employs **soft guidance** via energy minimization. This is a strategic design choice to balance reward maximization with physical plausibility, ensuring robustness against inevitable model biases.
>
> 3.  **Inference Efficiency (Guidance vs. Inner-Loop Solvers):**
>     *   The referenced methods often employ heavy inner-loop optimization (e.g., projection steps or augmented Lagrangian methods) at each denoising step to guarantee strict feasibility. This incurs significant computational overhead.
>     *   In contrast, DADP leverages the differentiable latent space of our VAEs to provide efficient, gradient-based guidance, thereby avoiding the significant latency associated with iterative inner-loop solvers.
>
> **Action:**
>
> We have expanded **Section 2 (Related Works)** to explicitly discuss these methods. We now classify them based on their constraint mechanisms and task scopes, clarifying that while they offer rigorous solutions for imitation and safe planning, DADP provides a specialized framework tailored for the **robustness and efficiency** requirements of Offline RL.

---

> ### Author Response · Authors · 2025-12-22
> **Response to Reviewer rjDr (Part 2/3)**
>
> ### 3. Usage of Uncertainty and $\gamma_t$
>
> **Reviewer Comment:** *Guidance weights do not explicitly incorporate model uncertainty... $\gamma_t$ is fixed to 1.*
>
> **Response:**
>
> We thank the Reviewer for this observation and clarify the role of uncertainty and the design choice of a fixed $\gamma_t$.
>
> 1.  **Implicit Uncertainty Handling via Manifold Learning:**
>     The primary motivation for using a probabilistic VAE is not merely to quantify output variance, but to learn a **structured, continuous latent manifold**. In Offline RL, deterministic dynamics models often produce sharp, erratic gradients in OOD regions. By contrast, our VAE ensures the gradient field remains informative and well-behaved. This structural regularization effectively handles the "uncertainty" by preventing the planner from exploiting adversarial gradients, achieving our goal without needing explicit variance-based weighting.
>
> 2.  **Robustness over Complexity:**
>     Empirically, DADP is robust to the choice of $\gamma_t$. With the regularized latent space, a simple fixed setting ($\gamma_t=1$) consistently performs well across benchmarks, avoiding additional hyperparameter tuning.
>
> 3.  **Potential for Adaptive Guidance:**
>     That said, we agree with the Reviewer that explicitly coupling $\gamma_t$ with the model's epistemic uncertainty (e.g., down-weighting guidance in highly uncertain regions) is a theoretically sound extension. While not necessary for the current benchmarks, it could further enhance safety in extreme OOD scenarios.
>
> **Action:**
>
> *   **Clarification:** In **Section 3.4**, we have added a discussion explaining that the **regularity of the VAE latent space** implicitly mitigates the risks of OOD planning, justifying the sufficiency of a fixed guidance weight.
> *   **Future Work:** We have updated **Section 5** to discuss uncertainty-adaptive guidance as future work.
>
> ### 4. Clarification on "Seeds"
>
> **Reviewer Comment:** *The reported 100-150 random seeds is unusually high for D4RL evaluations.*
>
> **Response:**
>
> We appreciate the opportunity to clarify the terminology used in our experimental setup.
>
> The reported numbers refer to the **total number of evaluation episodes**, not independent training runs. Due to stochastic diffusion sampling, multiple rollouts are required for stable estimation.
>
> Our protocol is consistent with with the standard benchmark settings established by the baseline *Diffuser* (Janner et al., 2022):
>
> 1.  **Training Seeds:** We trained all models using **3 independent random seeds**.
> 2.  **Evaluation Protocol:**
>     *   **Maze2D:** We aggregated results over **150 total episodes** (50 rollouts per training seed) to ensure robust estimation.
>     *   **MuJoCo:** We followed the standard protocol of **100 total episodes** (aggregated across seeds).
>
> This distinction ensures our reported metrics accurately reflect the robustness of the planner while remaining consistent with prior literature.
>
> **Action:**
>
> We have updated **Appendix A.3** and relevant **Table Captions** to explicitly distinguish between "Training Seeds" ($N=3$) and "Total Evaluation Episodes" ($M=100/150$) to prevent any future ambiguity.
>
> ### 5. Dynamics Model Validation & Metric Definition
>
> **Reviewer Comment:** *No analysis of dynamics model accuracy... The “dynamics inconsistency” metric is not fully operationalized.*
>
> **Response:**
>
> We agree that explicit validation strengthens the paper.
>
> 1.  **Reliability of Dynamics Models:**
>     We confirm that our underlying dynamics models are highly accurate. To substantiate this, we have compiled the **Held-out Test MSE** for both Forward and Inverse dynamics across all environments. The low prediction error confirms that the learned models capture the physics faithfully, ensuring that the gradient guidance provided to the planner is reliable and informative.
>
> 2.  **Rigor of Evaluation Metrics:**
>     We also appreciate the opportunity to clarify our evaluation protocol. It is important to distinguish between the *optimization objective* (which uses the learned model) and the *evaluation metric* (which must be objective). We confirm that all "dynamics inconsistency" results reported in the paper are calculated against the **ground-truth simulator**, providing an unbiased assessment of physical plausibility.
>
> **Action:**
>
> *   **Empirical Validation:** We have added **Appendix D**, which presents the detailed error analysis (MSE) for the learned dynamics models. The results demonstrate that our models achieve high predictive accuracy on unseen test data.
> *   **Metric Formalization:** We have revised **Section 3.3** and the **Figure 3 Caption** to explicitly define $E_{\text{eval}}$. We now formally state that while the planner optimizes against the learned dynamics energy, the reported inconsistency metric is the Euclidean distance between the planned transition and the **simulator's ground-truth next state**, ensuring a rigorous benchmark.

---

> ### Author Response · Authors · 2025-12-22
> **Response to Reviewer rjDr (Part 3/3)**
>
> ### 6. Combination of Variants & Return Predictor Details
>
> **Reviewer Comment:** *Does not evaluate their combination... Details about return predictor training are not fully specified.*
>
> **Response:**
>
> We thank the Reviewer for these suggestions regarding model design and reproducibility.
>
> 1.  **Design Philosophy on Combining Variants:**
>     While combining Forward (DADP-F) and Inverse (DADP-I) guidance is theoretically possible, we intentionally kept them distinct to prioritize **inference efficiency** and **ease of tuning**.
>     *   **Computational Cost:** Activating both streams simultaneously would double the gradient computation overhead at each denoising step, significantly increasing latency.
>     *   **Optimization Landscape:** Joint optimization introduces complex gradient interference between the forward and inverse objectives.
>     *   **Conclusion:** Our results show that a single, well-chosen guidance stream is sufficient to achieve state-of-the-art performance. We believe that maintaining a decoupled architecture offers a better trade-off for practical deployment than the marginal gains potentially offered by a heavier, combined model.
>
> 2.  **Reproducibility of the Return Predictor:**
>     We agree that full specification is essential for replication. The Return Predictor shares the same encoder backbone as the diffusion model to ensure feature consistency, a detail we have now explicitly documented.
>
> **Action:**
>
> *   **Discussion on Trade-offs:** We have added a discussion in **Section 4.3** clarifying that while combining variants is a valid extension, we advocate for the decoupled approach to maintain **computational parsimony** and avoid hyperparameter complexity.
> *   **Implementation Details:** We have updated **Appendix B** and **C** to include the exact architecture of the Return Predictor (shared backbone) and the training specifics (including the use of a **GaussianNormalizer** for target stability), ensuring the method is fully reproducible.
>
> ### 7. Broader Impact Statement
>
> **Reviewer Comment:** *A Broader Impact Statement should explicitly address safety risks and energy consumption.*
>
> **Response:**
>
> We share the Reviewer’s commitment to responsible AI research. We recognize that as generative planning methods advance toward real-world deployment, it is imperative to transparently discuss their operational boundaries and environmental footprints.
>
> 1.  **Safety Implications:** We acknowledge that policies guided by *learned* dynamics inherently carry risks related to model bias. In safety-critical domains (e.g., robotics), "hallucinated" physics can lead to dangerous behaviors. This reinforces the importance of our work in improving dynamics consistency, but also highlights the need for external safeguards (e.g., shielding) in deployment.
> 2.  **Computational Cost:** We are transparent about the trade-off inherent in diffusion models: the high-fidelity planning comes at the cost of iterative inference, which incurs a larger energy footprint than single-step policies.
>
> **Action:**
>
> We have included a formal **Broader Impact Statement** following the Conclusion. This section explicitly discusses:
>
> *   **deployment risks** associated with dynamics approximation errors.
> *   The **energy consumption** of iterative sampling, positioning it as a key motivation for future work on efficient inference techniques (e.g., consistency distillation).
>
> ---
>
> We thank the Reviewer again for the thorough and insightful evaluation. We have systematically addressed all raised concerns—clarifying our terminology, contextualizing our contributions against related works, and rigorously verifying our experimental protocols.
>
> We are confident that these revisions significantly strengthen the manuscript and look forward to the potential acceptance of our work.
>
> Sincerely,
>
> The Authors

---

### Review · Reviewer_pnnn · 2025-12-05

**Summary Of Contributions:**

The paper introduces Dynamics-Aligned Diffusion Planning (DADP) for offline RL: during each reverse-diffusion step, the denoising mean is guided by (i) a return objective and (ii) a dynamics-consistency objective so that sampled trajectories are both high-return and physically feasible. Two complementary variants are proposed: DADP-F uses a learned forward dynamics model to enforce state reachability; DADP-I uses a learned inverse dynamics model to enforce action executability. A unified gradient-guidance update and an explicit algorithm are provided.

On D4RL Maze2D/Multi-task and MuJoCo benchmarks, both variants outperform diffusion-based and standard offline RL baselines. Ablations show that removing dynamics alignment hurts performance; sensitivity analyses suggest robustness to guidance weights; an open-loop multi-step execution study illustrates a clear latency–performance trade-off.

**strengths**

1. Targets a real failure mode of trajectory diffusion (plan/execute mismatch) via differentiable dynamics alignment inside the sampler.

2. Clean, unified formulation (return + consistency in one step) with implementation details that appear reproducible.

3. Broad empirical support within D4RL, with ablations and sensitivity plots.

**weaknesses**

1. Relies on learned dynamics; under distribution shift, model bias can steer sampling toward infeasible regions.

2. Training/inference cost is non-trivial; evaluation is mostly simulator-based (no pixel observations, partial observability, or contact-rich tasks).

**Audience:**

Yes

**Audience Explanation:**

Readers in offline RL, diffusion-based planning, and guided generative sampling will find the “return + dynamics” in-sampler guidance useful. Practitioners concerned with planning latency will also be interested in the open-loop execution results and the practical guidance on when to prefer forward vs. inverse alignment.

**Broader Impact Concerns:**

We do not foresee specific ethical implications beyond standard considerations for offline RL research.

**Claims And Evidence:**

Yes

**Claims Explanation:**

The paper backs its central claims with competitive scores on multiple D4RL tasks, clear ablations (showing the benefit of alignment), and sensitivity/latency studies. The method is mathematically specified and algorithmically explicit.

**Requested Changes:**

**Critical**

1. Current evidence is on state-based, simulator-style D4RL settings. Author can add some sentences explicitly limiting claims to state-based offline RL and outlining broader settings (pixels/POMDP/contact-rich tasks) as future work.

**Nice-to-have**

2. A small efficiency–accuracy study makes the method easier to adopt. Can author provide a small study with reduced sampling steps to chart accuracy–latency trade-offs?

---

> ### Author Response · Authors · 2025-12-22
> **Response to Reviewer pnnn(Part 1/2)**
>
> **Dear Reviewer pnnn,**
>
> We thank you for your positive assessment of our work and for recognizing the clarity of the DADP formulation. We appreciate your constructive suggestions regarding the scoping of our claims and the evaluation of efficiency trade-offs.
>
> We have refined the manuscript to ensure our claims are precisely aligned with our empirical evidence. Below, we detail the revisions.
>
> ---
>
> ### 1. Scoping Claims to State-Based Settings (Critical)
>
> **Reviewer Comment:** *Current evidence is on state-based... Add sentences explicitly limiting claims... and outlining broader settings (pixels/POMDP) as future work.*
>
> **Response:**
>
> We agree with the Reviewer that precision regarding the problem setting is essential for scientific rigor.
>
> While the core principles of DADP (energy-guided diffusion) are theoretically generalizable, our current empirical investigation focuses on **state-based** control to isolate and validate the effects of dynamics consistency without the confounding factors of representation learning. We agree that explicitly bounding the scope to fully observable MDPs ensures that our claims are strictly commensurate with the presented evidence.
>
> Furthermore, we acknowledge that extending this framework to high-dimensional (Pixel) or partially observable (POMDP) settings involves addressing distinct challenges—such as learning robust latent dynamics—which we now categorize as a specific direction for future research.
>
> **Action:**
>
> *   **Precise Scope Definition:** We have revised the **Abstract** and **Introduction** to explicitly qualify our contributions. We now state that our experimental validation is conducted on **"state-based continuous control benchmarks,"** preventing any ambiguity regarding the method's immediate application domain.
> *   **Expanded Future Work:** We have updated the **Limitations** section (Section 5) to transparently discuss the extension to visual and partial-observability domains. We outline that future iterations of DADP could integrate **Latent World Models** to handle high-dimensional observations, positioning this as a promising avenue for subsequent research.
>
> ### 2. Efficiency-Accuracy Study (Nice-to-have)
>
> **Reviewer Comment:** *Can author provide a small study with reduced sampling steps to chart accuracy–latency trade-offs?*
>
> **Response:**
>
> We thank the Reviewer for this practical suggestion. Understanding the trade-off between inference latency and planning fidelity is indeed crucial for real-time deployment.
>
> We addressed this trade-off by analyzing two potential "control knobs": reducing the diffusion sampling steps ($N$) and amortizing inference via multi-step execution.
>
> 1.  **Sampling Steps ($N$) vs. Manifold Adherence:**
>     Our framework currently uses a standard DDPM scheduler. Unlike consistency models or DDIM, standard DDPM relies on a fixed noise schedule to map the Gaussian prior to the data distribution. We note that arbitrarily truncating $N$ (without retraining) creates a distribution mismatch that severely degrades the physical plausibility of trajectories. Given that our base schedule is already compact ($N=20$ for MuJoCo), further reduction yields diminishing returns in latency while risking significant performance collapse.
>
> 2.  **A Superior Trade-off: Open-Loop Execution:**
>     Instead of sacrificing generation quality, we identify **Open-Loop Multi-Step Execution** (as detailed in **Section 4.4**) as the more effective lever for this trade-off in robotic planning. By executing the first $k$ actions of a single high-quality plan, we **amortize** the inference cost over multiple environment steps.
>     *   **Evidence:** As shown in **Figure 6**, increasing the execution horizon significantly reduces the *effective latency per step* while maintaining high reward performance. This allows for a flexible "accuracy-latency" adjustment that is robust to the underlying generative process.
>
> **Action:**
>
> We have added a comprehensive discussion in **Appendix E** titled **"Efficiency Analysis: Sampling Steps vs. Execution Horizon."** This section:
>
> *   Formalizes the comparison between reducing $N$ and increasing execution horizon $k$.
> *   Synthesizes the results from Section 4.4 to explicitly answer the Reviewer's query, recommending Open-Loop Execution as the preferred strategy for balancing real-time constraints with planning accuracy.

---

> ### Author Response · Authors · 2025-12-22
> **Response to Reviewer pnnn(Part 2/2)**
>
> ### 3. Model Bias and Broader Impact
>
> **Reviewer Comment:** *Relies on learned dynamics... model bias can steer sampling.* / *No specific ethical implications.*
>
> **Response:**
>
> *   **On Model Bias:** We agree with the Reviewer that reliance on learned dynamics is an **inherent characteristic** of our approach. Inaccuracies in the dynamics model (epistemic uncertainty) can indeed provide erroneous guidance, particularly in OOD regions. However, we emphasize that DADP uses dynamics as *soft guidance* on top of a strong data-driven prior (the diffusion model), which provides inherent regularization against model exploitation compared to pure model-based planning. Nevertheless, we agree that explicitly documenting this dependency is crucial for potential users.
> *   **On Broader Impact:** We share the Reviewer's view that responsible AI research requires looking beyond immediate ethical violations to consider downstream effects. We believe discussing the operational risks (safety) and environmental costs (energy) contributes to the maturity of the field.
>
> **Action:**
>
> *   **Explicit Risk Assessment:** We have updated **Section 5 (Limitations)** to transparently discuss the "Dependency on Model Fidelity." We clarify that while DADP is robust, its performance is ultimately bounded by the dynamics model's accuracy, and severe distribution shifts may necessitate additional uncertainty quantification mechanisms.
> *   **Broader Impact Statement:** We have included a dedicated section following the Conclusion, addressing:
>     1.  **Deployment Safety:** The risks of "hallucinated physics" in safety-critical domains (e.g., robotics), advocating for the use of external safeguards (shielding).
>     2.  **Sustainability:** The energy footprint of iterative diffusion inference, framing it as a key motivation for our ongoing work on efficient open-loop execution.
>
> ---
>
> We thank the Reviewer again for the encouraging assessment and the insightful suggestions regarding scope and efficiency. We have systematically addressed the requested changes—precisely bounding our claims to state-based settings, providing a deeper analysis of the accuracy-latency trade-off, and transparently discussing the method's limitations.
>
> We are confident that these revisions have sharpened the contribution of the manuscript, and we look forward to the potential acceptance of our work.
>
> Sincerely,
>
> The Authors

---

### Review · Reviewer_zZvq · 2025-12-09

**Summary Of Contributions:**

## Summary

This paper investigates diffusion-based planning for offline reinforcement learning, highlighting a key failure mode: trajectories generated by a diffusion model can yield high rewards under a learned reward model but may be inconsistent with the true environment dynamics, resulting in significant discrepancies between planned and executed rollouts. To address this, the authors propose Dynamics-Aligned Diffusion Planning (DADP), which augments standard reward-guided diffusion planning with an explicit dynamics-alignment energy term derived from either a forward or inverse dynamics model trained from offline data. At each denoising step, the trajectory is updated using the gradient of a joint objective that increases predicted return while penalizing discrepancies between predicted and dynamics-consistent transitions. The method is instantiated in two variants: DADP-F (forward dynamics alignment) and DADP-I (inverse dynamics alignment), both of which are implemented via VAE-based dynamics models. Experiments on the D4RL Maze2D and MuJoCo benchmarks demonstrate that DADP enhances both planning performance and robustness to open-loop execution, outperforming prior diffusion-based planners, such as Diffuser and HDMI, and competing with strong offline RL baselines.

**Audience:**

Yes

**Audience Explanation:**

TMLR’s audience includes researchers in reinforcement learning, generative modeling, and decision-making under uncertainty, all of whom are likely to be following the recent trend of using diffusion models for planning and control.

**Claims And Evidence:**

Yes

**Claims Explanation:**

The central empirical claim that incorporating explicit dynamics alignment into reward-guided diffusion planning improves performance and open-loop robustness relative to reward-only diffusion guidance is well supported by the reported results on D4RL Maze2D and MuJoCo benchmarks, including baseline comparisons, ablations, and sensitivity analyses. The paper provides evidence that DADP reduces a carefully defined measure of dynamics inconsistency and that this reduction correlates with improved return under open-loop execution in the tested environments. However, some of the broader claims about the general mitigation of compounding errors and their applicability to more complex domains are currently supported only indirectly by these experiments and heuristic arguments; as such, they should be interpreted as promising indications rather than fully general guarantees. Within the scope of the benchmarks and settings actually evaluated, the evidence is accurate and clearly presented.

**Requested Changes:**

- **Clarify the "Theoretical Insight" Contribution:** In Section 1 (Introduction), contribution #3 claims "We provide an analysis showing that explicit dynamics alignment mitigates compounding errors..." However, the paper does not appear to contain a formal theoretical analysis or derivation bounding the compounding errors. Section 3 presents the formulation of the method and metrics, but not a theoretical proof or a rigorous analytical bound. Please either add a formal theoretical analysis (e.g., a lemma bounding the divergence between planned and executed trajectories given the alignment term) or rephrase the contribution claim to reflect that the insight is empirical or architectural (e.g., "We propose a formulation that explicitly targets dynamics consistency..."). Mischaracterizing a heuristic formulation as "theoretical analysis" should be corrected.

- **Computational Cost Quantification:** The method requires computing gradients through a VAE (forward or inverse) at _every_ denoising step. This likely adds significant wall-clock overhead compared to the base Diffuser model. While Figure 6 discusses rollout time versus performance for open-loop settings, there is no direct comparison of inference latency per planning step between Diffuser and DADP. Please provide a table or text report detailing the average wall-clock time required to generate a single plan (or a batch of plans) for Diffuser, DADP-F, and DADP-I on the same hardware.

- **Dynamics Model Quality Analysis:** The success of DADP relies on the accuracy of the learned dynamics VAEs. If the dynamics model is inaccurate, the alignment term would guide the planner away from feasible regions. Please include a brief evaluation of the dynamics models themselves. For example, report the reconstruction error (MSE) of the Forward/Inverse VAEs on a held-out test set for the different datasets. This will help clarify if the performance gains correlate with the quality of the learned dynamics.

---

> ### Author Response · Authors · 2025-12-22
> **Response to Reviewer zZvq**
>
> **Dear Reviewer zZvq,**
>
> Thank you for the positive assessment and for recognizing our empirical results on the D4RL and MuJoCo benchmarks. We appreciate the constructive feedback regarding terminology precision and component validation. We have revised the manuscript accordingly, as detailed below.
>
> ------
>
> ### 1. Clarification of "Theoretical Insight"
>
> **Reviewer Comment:** *Clarify the "Theoretical Insight" Contribution... The paper does not appear to contain a formal theoretical analysis... Mischaracterizing a heuristic formulation as "theoretical analysis" should be corrected.*
>
> **Response:**
>
> We appreciate the emphasis on terminological rigor. We acknowledge that the manuscript does not present formal convergence theorems, which may render the term "Theoretical Insight" ambiguous. However, we emphasize that our approach is **not heuristic**.
>
> We provide a **Principled Formulation** derived from first principles. Specifically, in **Section 3.5**, we mathematically formulate dynamics inconsistency as an energy potential field and derive the exact guidance gradient $\nabla_{\mu} (\lambda \mathcal{J} - \beta E)$ that projects the diffusion process onto the learned manifold. This formulation offers a structural explanation—a **mechanistic understanding**—of why DADP eliminates "hallucinations."
>
> However, to align with strict mathematical conventions, we are happy to adopt more precise terminology.
>
> **Action:**
>
> *   **Refined Terminology:** We have updated the **Introduction (Contribution List)** and section headers, replacing "Theoretical Insight" with **"Principled Formulation and Mechanistic Analysis."**
> *   **Reframing Claims:** We have rephrased the narrative to emphasize that our approach is **mathematically grounded** in energy-based guidance. This ensures the claims accurately reflect the analytical nature of our derivation (Eq. 14-16) while avoiding the implication of formal convergence proofs.
>
> ### 2. Computational Cost Quantification
>
> **Reviewer Comment:** *This likely adds significant wall-clock overhead... Please provide a table or text report detailing the average wall-clock time required to generate a single plan.*
>
> **Response:**
>
> We thank the Reviewer for raising the critical issue of real-world deployment efficiency.
>
> We are pleased to report that contrary to the concern of significant overhead, DADP is designed to be highly efficient. The structural key is that our learned dynamics model (simple MLP-based VAE) is orders of magnitude more lightweight than the diffusion backbone (U-Net). Consequently, the additional backward pass for guidance gradients incurs a **negligible computational cost** relative to the expensive denoising iterations.
>
> **Action:**
>
> *   **New Benchmarking Section:** We have added **Appendix E: Inference Latency Analysis**, providing a rigorous wall-clock time comparison.
> *   **Quantitative Evidence (Table 5):** We introduced a new **Table 5**, reporting the average seconds-per-plan for Diffuser, DADP-F, and DADP-I over 100 trials.
> *   **Key Result:** The data explicitly validates our efficiency claims, showing that DADP incurs only a **marginal latency overhead (~4-5%)** compared to the unguided baseline. This confirms that our framework effectively achieves dynamics consistency without compromising the real-time feasibility required for control tasks.
>
> ### 3. Dynamics Model Quality Analysis
>
> **Reviewer Comment:** *Please include a brief evaluation of the dynamics models themselves... report the reconstruction error (MSE).*
>
> **Response:**
>
> We appreciate the opportunity to substantiate the fidelity of our underlying components.
>
> We agree that verifying the predictive accuracy of the dynamics model is crucial. Since DADP relies on differentiating through this model, demonstrating low prediction error serves as the primary validation that the generated gradients are physically informative and not merely optimizing against model artifacts.
>
> **Action:**
>
> *   **Empirical Validation:** We have introduced a new **Appendix D: Dynamics Model Accuracy**.
> *   **Quantitative Evidence (Table 4):** We explicitly report the **Held-out Test MSE** for both Forward (state transition) and Inverse (action reconstruction) models across all benchmark environments.
> *   **Conclusion:** The results demonstrate **consistently low prediction errors** on unseen data. This confirms that the learned models effectively capture the environment's physical laws, ensuring that the gradient guidance provided to the diffusion planner is both accurate and robust.
>
> ------
>
> We thank the Reviewer again for the constructive feedback. These revisions clarify our terminology, quantify computational overhead, and rigorously validate the dynamics backbone, substantially strengthening the manuscript.
>
> Sincerely,
>
> The Authors

---

### Author Response · Authors · 2025-12-22
**General Response to All Reviewers**

**Dear Action Editor and Reviewers,**

We thank you for your time and the comprehensive feedback provided on our manuscript. We appreciate the consensus among reviewers regarding the **novelty of our unified framework (DADP)** and the **strong empirical performance** demonstrated across the D4RL benchmarks.

We have leveraged this review process to significantly enhance the rigor, clarity, and transparency of our work. Based on your constructive comments, we have executed a major revision. The key improvements are summarized below:

1.  **Refining Methodological Scope & Terminology (Addressing R1, R3):**
    *   **Terminological Precision:** We have refined the characterization of our contribution from "Theoretical Insight" to **"Principled Formulation and Mechanistic Analysis."** This terminology accurately reflects that our method is not a heuristic, but is mathematically derived as a gradient-based energy minimization process (Eq. 16) that projects trajectories onto the dynamics manifold.
    *   **Contextual Positioning:** We have expanded **Section 2** to clearly distinguish DADP (Offline RL/Stitching) from hard-constraint methods (Imitation/Control). We emphasize that our "soft guidance" approach is a deliberate design choice for robustness against model uncertainty.
    *   **Protocol Clarity:** We resolved the ambiguity regarding experimental protocols, explicitly distinguishing between training seeds ($N=3$) and total evaluation episodes ($M=100/150$) to ensure full alignment with standard benchmarks.

2.  **Quantification of Efficiency & Overhead (Addressing R2, R3):**
    To address inquiries regarding real-world deployment, we have added **Appendix E** featuring a rigorous **Inference Latency Analysis**.
    *   **Key Finding:** New benchmarks (**Table 5**) reveal that DADP incurs only a **marginal wall-clock overhead (~4-5%)** compared to the unguided baseline.
    *   **Structural Reason:** We clarify that this efficiency stems from the lightweight architecture of our VAE dynamics model compared to the diffusion backbone.

3.  **Validation of Core Components (Addressing R1, R3):**
    To substantiate the reliability of our guidance gradients, we have introduced **Appendix D**, which reports the held-out **Test MSE** for both forward and inverse dynamics models (**Table 4**). The consistently low prediction errors confirm that the learned physics models are accurate and robust.

4.  **Responsible Scoping & Broader Impact (Addressing R2, All):**
    *   **Explicit Scoping:** We have precisely bounded our claims to **state-based continuous control**, categorizing high-dimensional pixel-based extensions as future work to maintain scientific rigor.
    *   **Impact Statement:** We have added a dedicated **Broader Impact Statement** to transparently discuss the operational risks (e.g., hallucinated physics in safety-critical tasks) and the environmental footprint of iterative diffusion planning.

We believe these revisions comprehensively address the raised concerns and significantly strengthen the scientific quality of the manuscript. We have provided detailed, point-by-point responses to each reviewer below.

Sincerely,

The Authors

---

### Decision · Action_Editor_Rt4u · 2026-01-25

**Recommendation:** Accept with minor revision

**Additional Comments:**

1. In the review process, this work adds additional discussion on the DPCC method instead of the additional experiment to address the concerns of the reviewer. It would be helpful to add experiments to compare DADP and DPCC in the state-based RL.

**Audience:**

Yes

**Audience Explanation:**

The paper studies an important dynamics inconsistency problem in state-based offline reinforcement learning and diffusion-based planning. As decision-making and robotics usually aim to generate a valid action to achieve a valid state, the DADP framework offers a solution in the domain. The insights regarding the trade-off between forward (state-level) and inverse (action-level) guidance are also valuable for practitioners.

**Claims And Evidence:**

Yes

**Claims Explanation:**

This work studies the diffusion-based planning on the state-based offline RL and proposes the dynamics-aligned Diffusion Planning (DADP) with a dynamics-alignment energy term to force the physical consistency. There are two variants, stated-level DADP (DADP-F) and action-level DADP (DADP-I), which have shown a great performance in the state-based D4RL environment.

Accurate and Clear Evidence

1.	This work has shown the performance of DADP-F and DADP-I on state-based D4RL Maze2D and MuJoCo benchmar,k and outperforms baselines such as Diffuser and HDMI.

2.	During the discussion process, this work also further provides the computation costs and test MSE loss to address the concerns of reviewers. More specifically, they show that the wall-clock latency of DADP overhead is marginal, 5% and that the learned dynamics models achieve low Mean Squared Error on held-out test sets.

3.	For the presentation, this work also uses the "principled formulation" instead of "theoretical insight", which is more accurate.

---

> ### Author Response · Authors · 2026-02-20
> **Author Response to Minor Revision and Camera-Ready Submission (Paper ID: 6468)**
>
> **Dear Action Editor,**
>
> We are pleased to submit the final camera-ready version of our manuscript, **"Dynamics-Aligned Diffusion Planning for Offline RL: A Unified Framework with Forward and Inverse Guidance."** We have addressed all the minor revision requirements as follows:
>
> 1. **Baseline Comparison with DPCC:** As requested, we have incorporated a comparative study with the DPCC method on the **Maze2D** benchmark (Table 1). We have also added a technical discussion in **Section 4.1** comparing our soft guidance approach with DPCC's hard projection, explaining why DADP is more effective for stitching behaviors from suboptimal datasets.
> 2. **Terminology Update:** We have replaced the phrase "theoretical insight" with "**principled formulation**" throughout the manuscript to more accurately describe our contribution.
> 3. **Efficiency and Accuracy Metrics:** We have added **Appendix E** and **Appendix F** to report the wall-clock latency analysis and the dynamics model test Mean Squared Error (MSE). The results demonstrate that DADP incurs marginal computational overhead  and that our VAE-based dynamics models achieve high predictive accuracy.
> 4. **Final Formatting:** The manuscript has been de-anonymized to include author names, affiliations, and funding acknowledgments. All revision-specific coloring has been removed.
>
> **Code Availability:**
> The official implementation of DADP is now publicly available at: https://github.com/Wzhhhh0815/Dynamics-Aligned-Diffusion-Planning
>
> We thank the Action Editor and the reviewers for their constructive feedback, which has significantly improved the quality of our work.
>
> **Best regards,**
>
> **Zihao Wang**